# Evaluation of VIIRS Neural Network Cloud Detection against Current Operational Cloud Masks

Charles H. White[1], Andrew K. Heidinger[2], and Steven A. Ackerman[1]

[1]University of Wisconsin - Madison, Department of Atmospheric and Oceanic Sciences, Madison, WI, USA
[2]NOAA/NESDIS/Center for Satellite Applications and Research, Madison, WI, USA

**Correspondence:** Charles H. White (cwhite25@wisc.edu)

**Abstract.**

Cloud properties are critical to our understanding of weather and climate variability, but their estimation from satellite imagers is a nontrivial task. In this work, we aim to improve cloud detection which is the most fundamental cloud property. We use a neural network applied to Visible Infrared Imaging Radiometer Suite (VIIRS) measurements to determine whether an imager pixel is cloudy or cloud-free. The neural network is trained and evaluated using four years (2016-2019) of coincident measurements between VIIRS and the Cloud-Aerosol Lidar with Orthogonal Polarization (CALIOP). We successfully address the lack of sun glint in the collocation dataset with a simple semi-supervised learning approach. The results of the neural network are then compared with two operational cloud masks: the Continuity MODIS-VIIRS Cloud Mask (MVCM) and the NOAA Enterprise Cloud Mask (ECM).

We find that the neural network outperforms both operational cloud masks in most conditions examined with a few exceptions. The largest improvements we observe occur during the night over snow or ice covered surfaces in the high latitudes. In our analysis, we show that this improvement is not solely due to differences in optical depth-based definitions of a cloud between each mask. We also analyze the differences in true positive rate between day/night and land/water scenes as a function of optical depth. Such differences are a contributor to spatial artifacts in cloud masking and we find that the neural network is the most consistent in cloud detection with respect to optical depth across these conditions. A regional analysis over Greenland illustrates the impact of such differences and shows that they can result in mean cloud fractions with very different spatial and temporal characteristics.

## 1 Introduction

Clouds serve many critical roles in the earth's weather and climate system, and are one of the largest sources of uncertainty in future climate scenarios (Stocker et al., 2013). Determining their presence in current observational records is a fundamental first step in understanding their variability and impact. Polar-orbiting satellite imagers such as the Visible Infrared Imaging Radiometer Suite (VIIRS; Cao et al., 2013) offer frequent views of global cloud cover at high spatial resolution. However, cloud detection from passive visible and infrared observations is a nontrivial problem. This is particularly true for clouds with low optical depths, and clouds above cold and visibly reflective surfaces (Ackerman et al., 2008; Holz et al., 2008). These

qualifications on imager cloud detection make it difficult to construct confident observational analyses of cloud variability from passive satellite instruments especially in the polar regions. As a result, many differences exist between cloud climate records made with different algorithms, or sensors with different capabilities (Stubenrauch et al., 2013).

Machine learning (ML) has become a popular tool for statistical modeling in earth sciences including the use of both supervised and unsupervised methods. Supervised ML methods in the earth sciences can require large amounts of training data often created from physically-based models, obtained from manual labeling, or observed from other instrument platforms. These approaches have been extensively used in characterizing the surface and atmosphere from remote sensing instruments. A sample of popular ML approaches (and their applications) used in satellite meteorology include naïve bayesian classifiers (Uddstrom et al., 1999; Heidinger et al., 2012; Cintineo et al., 2014; Bulgin et al., 2018), random forests (Kühnlein et al., 2014; Thampi et al., 2017; Wang et al., 2020), and neural networks (Minnis et al., 2016; Håkansson et al., 2018; Sus et al., 2018; Wimmers et al., 2019; Marais et al., 2020).

In this analysis, we develop a neural network cloud mask (NNCM) that uses the moderate resolution channels from VIIRS to determine whether a given imager pixel contains a cloud or is cloud-free. We train the neural network using observations from the Cloud-Aerosol Lidar with Orthogonal Polarization (CALIOP; Winker et al., 2009). Observations from CALIOP are often used to validate cloud masks and cloud property estimates due to the instrument's ability to retrieve vertical profiles of the atmosphere and characterize clouds with low optical depth. Additionally, its placement in the A-train constellation makes it a convenient reference for Moderate Resolution Imaging Spectroradiometer (MODIS) cloud property validation (Holz et al., 2008). The Suomi National Polar-orbiting Partnership (SNPP) VIIRS instrument, despite not being in the A-train constellation, makes spatially and temporally coincident observations with CALIOP roughly every two days. Thus, there is opportunity for matching observations between these two sensors with some limitations. One such limitation is that the range of atmospheric and surface conditions sampled by CALIOP do not necessarily match that of SNPP-VIIRS. Conditions where collocations between these two sensors occur are even less representative, and do not contain instances of significant sun glint. In this work we demonstrate how a very simple semi-supervised learning approach can ameliorate this specific limitation.

There are several recent applications of ML in characterizing clouds from imager observations that use CALIOP as a source of labeled data. Perhaps most relevant is Wang et al. (2020) in which several random forest (RF) models are trained to identify the presence and phase of clouds from VIIRS observations under somewhat idealized conditions (spatially homogeneous and low aerosol optical depths). In such conditions the, RF models demonstrated improvements in cloud masking and cloud phase determination over current algorithms. Håkansson et al. (2018) uses CALIOP as a training source for estimating MODIS cloud-top heights with precomputed spatial features, MODIS brightness temperatures, and numerical weather prediction (NWP) temperature profiles using a neural network. They additionally demonstrate the ability to accurately estimate cloud-top heights with channels only available on sensors such as the Advanced Very High Resolution Radiometer (AVHRR) and VIIRS. Similarly, Kox et al. (2014) trained a neural network with CALIOP to determine the presence of cirrus clouds and estimate their optical depth and cloud-top height from SEVIRI observations. The Community Cloud retrieval for CLimate (CC4CL; Sus et al., 2018) also uses neural network based approaches for imager cloud detection. The CC4CL neural network models are trained with collocations between the Advanced Very-High Resolution Radiometer (AVHRR) and CALIOP. Adjustments are applied to

shared MODIS and Advanced Along-Track Scanning Radiometer (AATSR) channels (accounting for differences in spectral response functions) to ensure the approaches generalize beyond AVHRR to those imagers as well. While the majority of these applications for cloud property estimates are relatively recent, there were successful implementations of ML approaches well before the launch of CALIOP using manually labeled scenes (Welch et al., 1992).

Our approach aims to improve upon existing literature in several ways. Significant effort has gone into determining useful spectral characteristics in the development of past imager cloud masks. Still, it is possible that not all relevant variability is being exploited particularly that which involves three or more channels. Rather than relying on precomputed spectral or textural features, we allow a neural network to learn relevant features from a local 3 pixel by 3 pixel image patch from all 16 moderate resolution VIIRS channels. This necessitates a relatively large neural network architecture in order to exploit the variability of these observations to discriminate cloudy from cloud-free scenes. We train the model without filtering CALIOP collocations to encourage more reliable predictions under non-ideal conditions. Additionally, we specifically address issues caused by the lack of sun glint scenes in collocations between SNPP VIIRS and CALIOP. This specific implementation does not require surface temperature, surface emissivity, the use of clear-sky radiative transfer modeling, snow cover, or ice cover information. The only ancillary data used is a VIIRS-derived land/water mask in the level-1 geolocation product. The NNCM uses a single model for all surface types and solar illumination conditions and in some respects, greatly simplifies the processing pipeline for imager cloud masking.

In this analysis, we demonstrate that a neural network cloud mask (NNCM) can outperform two operational VIIRS clouds masks in detecting clouds identified by CALIOP. In particular, we note large improvements at night in the middle and high latitudes. Since cloud masks may have differing definitions of what substantiates a cloud, we evaluate the performance of each approach after removing clouds above an increasing lower optical depth threshold. The usefulness of the predicted probabilities as a proxy for uncertainties are assessed. We also show an example of how differences in cloud detection ability can result in vastly different spatial and temporal characteristics of regional mean cloud cover assessments in the polar regions.

## 2 Instruments and Data

### 2.1 VIIRS

VIIRS is a polar-orbiting visible, near-infrared, and infrared imager on board the S-NPP and NOAA-20 satellites. The swath width of VIIRS is roughly 3060 km allowing for at least twice daily views of any given ground location and more frequent views at higher latitudes. VIIRS altogether measures top-of-atmosphere radiation for 22 different channels. This is made up of five imaging channels (I-bands) with a nadir resolution of 375 m, and sixteen moderate resolution channels (M-bands) with a nadir resolution of 750 m (Table 1). VIIRS has an additional Day/Night Band (DNB) for nocturnal low-light applications. This work is focused entirely on the sixteen moderate resolution channels and does not include the use of the higher resolution I-bands or the DNB. Furthermore, we only consider VIIRS data from S-NPP which has an equatorial crossing time of 1:30 pm.

## 2.2 CALIOP

CALIOP is polar-orbiting lidar taking near-nadir observations on board the Cloud-Aerosol Lidar and Infrared Pathfinder Satellite Observations (CALIPSO) satellite which also has an equatorial crossing time of roughly 1:30 pm. CALIOP measures at wavelengths of 1064 nm and 532 nm with a horizontal resolution of 333 m. The individual lidar footprints are aggregated in the creation of both the 1 km and 5 km CALIOP Cloud Layers products. CALIOP's ability to characterize optically thin cloud layers make it a suitable validation source for imager cloud masking. While CALIOP, in many respects, is the more appropriate instrument for accurately estimating cloud properties (including cloud detection), its spatial sampling is extremely sparse relative to VIIRS and other imagers. This motivates our goal of extending CALIOP's cloud detection ability to passive imager measurements.

## 2.3 MVCM and ECM

Current operational cloud masks for VIIRS include the NOAA Enterprise Cloud Mask (ECM; Heidinger et al., 2012; Heidinger et al., 2016), and the Continuity MODIS-VIIRS Cloud Mask (MVCM; Frey et al. 2020). The ECM algorithm was originally designed for AVHRR climate applications and has since been extended to a wide range of geostationary and polar-orbiting imagers including VIIRS. This approach is based on several naive bayesian classifiers that are each trained specifically for different surface types. This approach is similarly trained using CALIOP collocations with VIIRS and makes probabilistic predictions of cloudy or cloud-free pixels. A key advantage of the ECM's naive bayesian approach is that certain predictors can be removed or turned off (such as visible channels during the night). Due to the simplicity of naive bayesian classifiers, the ECM is overall more interpretable than our proposed neural network.

The MVCM has heritage with the MODIS cloud mask (Ackerman et al., 2010), and has been adjusted to only use channels available on both VIIRS and MODIS. Obtaining continuity in cloud detection between the two imagers is a specific goal of the MVCM. The MVCM has a collection of cloud tests each with specified low-confidence and high-confidence thresholds used in a fuzzy-logic approach. The specific tests that are applied are determined by solar illumination and the surface type. The clear-sky confidence values imparted by each applied test are combined to produce a preliminary overall clear-sky confidence value which can then be modified by clear-sky restoral tests. The MVCM's reliance on physically-based reasoning also make its predictions relatively interpretable compared to our neural network approach.

## 2.4 Collocation Methodology

The labeled data that is used to train and evaluate the performance of the neural network comes from version 4.2 of the 1 km CALIOP Cloud Layers product (Vaughan et al., 2009). A vertical profile is determined to be cloudy when the number of cloud layers is equal to or exceeds one. Otherwise the profile is assumed to be cloud-free. The CALIOP labels are set to zero for cloud-free observations, and one for cloudy observations. Other CALIOP information such as the cloud-top pressure and cloud feature type are used in the validation of the cloud masks. Cloud optical depth is obtained from the 5 km CALIOP Cloud Layers product since it is unavailable at the 1 km resolution. There are difficulties in matching satellite imager measurements

with CALIOP. Many of these issues are discussed at length in Holz et al. (2008), and include differences in spatial footprint, viewing angle, the observation time between the two instruments, and the horizontal averaging applied within the CALIOP products to increase their signal to noise ratio.

Collocations between SNPP VIIRS and CALIOP are obtained by performing a nearest neighbors search between the 1 km CALIOP Cloud Layers product, and the 750 m (at nadir) VIIRS observations. A parallax correction is then applied to account for pixels with high altitude clouds that are observed at oblique viewing angles by VIIRS. The details of the parallax correction are identical to that of Holz et al. (2008). Collocations with times that differ by more than 2.5 minutes are removed. This is a particularly strict requirement relative to Heidinger et al. (2016) which uses a limit of 10 minutes and severely limits both the number of possible collocations between these instruments and the range of viewing conditions sampled. We make this choice because the time difference between observations is a critical factor in the representativeness of a CALIOP profile for a given imager pixel. This is particularly true for small clouds that occupy a horizontal area similar to or smaller than a single VIIRS pixel in environments with high wind speeds. Collocations are found for these instruments from January 2016 through December 2019. Some gaps in the collocation dataset exist and are primarily due to the availability of CALIOP data products. Following the recommendations from the CALIPSO team, we remove all CALIOP profiles that contain low-energy laser shots with 532 nm laser energies less than 80 mJ. This results in a relative sparsity of collocations over central South America after mid-2017. In total, roughly 27.1 million collocations were collected for this study with the above requirements.

### 2.5 Neural Network Inputs

The observations used as input into the neural network come from the moderate resolution channels (M1-M16; Table 1) obtained from the NASA processing of SNPP VIIRS. All channels are either expressed as a reflectance or brightness temperature. In addition to the VIIRS channels we also include a binary land/water mask, solar zenith angle, sun glint zenith angle, and the absolute value of latitude. The binary land/water mask is created from an eight-category land/water mask included the in the VNP03MOD geolocation product which includes land, coastline, and various types of water surfaces. Our binary mask is created by grouping together all water surfaces as a single water category, and grouping together land and coastline as a single land category. Sun glint zenith angle is the angle between the surface normal of the estimated specular point (the point of maximum sun glint) and atmospheric path viewed by VIIRS. For each of the twenty inputs, a 3 pixel by 3 pixel array is extracted and is used to predict the cloudy or cloud-free label at the center pixel.

The VIIRS/CrIS fusion channels (Weisz et al., 2017) are estimates of MODIS-like channels using coarse-resolution measurements from the Cross-track Infrared Sounder (CrIS) that are interpolated to match the moderate resolution channels of VIIRS. A subset of the VIIRS/CrIS fusion channels without solar contributions (Table 2) are used in a pseudo-labeling model for sun glint scenes (described later in section 3.1), but these are not used in the final NNCM model. Table 3 summarizes which inputs are used for the NNCM, a neural network without pseudo-labeling, and the pseudo-labeling model.

## 2.6 Dataset Splitting

In statistical modeling it is important to ensure independence between the training, validation, and testing datasets. The CALIOP Cloud Layer product's feature identification algorithm often relies on horizontal averaging to detect cloud layers of low optical depth. This averaging increases the signal to noise ratio and allows for more accurate identification of such features. As a result, clouds with low optical depth may have their attributes replicated across neighboring CALIOP profiles. As pointed out in Håkansson et al. (2018), separating imager and CALIOP collocations by random sampling would result in three nearly identical datasets and would yield a model that greatly overfits. To avoid this, we stratify our collocations by year into our training set that consists of 14.3 million collocations from 2016 and 2018, a validation set consisting of 5.7 million collocations from 2017, and our testing set consisting of 7.1 million collocations from 2019. The training set is what is supplied to the model during the training stage. The validation dataset is used for hyperparameter tuning during model development and early stopping during the training stage. The testing set is used to provide estimates of model performance which we will analyze in section 4 and is not seen by the model during the training or hyperparameter tuning stages.

The spatial and seasonal distribution of these collocations can be seen in Fig 1. There are slight differences in spatial sampling between the testing dataset and the validation and training datasets. We expect that this is due to a combination of the strict 2-minute time difference we require of the collocations and the exit of CALIPSO from the A-train in late 2018 (Braun et al., 2019). We select 2019 for our testing dataset since it provides the most spatially and temporally complete dataset. 2016 and 2018 are used in our training dataset since they offer the next largest number of collocations. We judged that 2017 was the least spatially and temporally representative hence its use only as a validation dataset for hyperparameter tuning and early stopping during training.

## 2.7 CALIOP Data Preprocessing

A common preprocessing step when training imager cloud masks with CALIOP observations is to filter the collocations using several heuristics in order to infer when CALIOP cloud detection is unreliable or unrepresentative of the corresponding imager pixel. Heidinger et al. (2012) filters AVHRR collocations so that only CALIOP observations where the 5 km along-track cloud fraction is equal to 0% or 100% are included. Holz et al. (2008) only retained MODIS pixels where all collocated CALIOP retrievals are identical. Wang et al. (2020) require that both the 1 km and 5 km CALIOP Cloud Layer products agree, that five consecutive 1 km CALIOP profiles agree, and they additionally remove profiles with high aerosol optical depths. Many of these filters achieve a similar result in requiring that CALIOP profiles, to a varying degree, are spatially homogeneous with regards to the presence of clouds. This filtering is often applied to remove fractionally cloudy profiles or profiles where the clouds may have moved out of the corresponding imager pixel. Karlsson et al. (2020) employ an approach that filters AVHRR/CALIOP collocations on the basis of cloud optical depth. This is done in an iterative fashion in order to determine the lower optical depth threshold in which their cloud masking method can reliability detect clouds.

In our approach, we intentionally do not perform any of the above preprocessing steps to our training dataset. This is because we include a substantial amount of spatial information in our neural network inputs. If such a spatial filter were applied to the

CALIOP data, then cloud edges and small clouds (often boundary-layer clouds) would rarely occur in our training dataset. This would yield a large amount of bias in a model that accounts for any amount of spatial variability and could cause it to generalize poorly. Alternatively, we apply a spatial filter to only our testing dataset to create a second filtered testing dataset
that we can evaluate our models against. This allows us to evaluate the performance of our cloud masking model against others using only the most reliable CALIOP collocations without biasing any model that considers spatial variability. Additionally, we can analyze the performance of our neural network approach in fractionally cloudy scenes using the unfiltered testing dataset with the knowledge that these collocations may be overall less reliable. The specific filter we apply to our testing dataset requires that five consecutive 1 km profiles agree. This spatial filter creates a filtered testing dataset of 5.9 million collocations
compared to the unfiltered testing dataset of 7.1 million collocations. In no way does this filter affect the training or validation data.

## 3    Methods

### 3.1    Pseudo-Labeling Procedure

A general concern in using statistical models such as neural networks, is the ability for them to generalize to unseen data.
One such scenario in this dataset is sun glint. Sun glint is the specular reflection of visible light usually over water surfaces which results in very large visible reflectivity for both cloudy and cloud-free observations. In our dataset of VIIRS/CALIOP collocations, we never observe any substantial amount of sun glint. Thus, without accounting for sun glint, any statistical model will likely fail to make a reasonable assessment of cloud cover under these conditions. Often, this results in erroneously predicting cloud cover in sun glint regions due to their high visible reflectivity. In the ECM, sun glint is handled by turning off
cloud tests that use visible and shortwave infrared channels with solar contributions. In the MVCM, this is handled by decision paths that use visible channels to detect clear-sky pixels specifically in sun glint regions. CLDPROP optical properties (which use the MVCM) also use a clear-sky restoral algorithm (Platnick et al., 2017) in an attempt to remove erroneously cloudy pixels, but it is not included in the MVCM output.

We aim to overcome this limitation by using a simple semi-supervised learning approach called pseudo-labeling (Lee,
2013). Pseudo-labeling is the approach of using a model to make predictions on unlabeled data, assuming that some or all of these predictions are correct, and adding these predictions to the original training dataset as if they were true labels. In our application, the pseudo-labeling model only uses VIIRS and VIIRS/CrIS fusion channels unaffected by sun glint, and the final NNCM model uses all VIIRS channels and no VIIRS/CrIS fusion channels. Stated simply, adding these pseudo-labels to the training dataset incentivizes the final NNCM model to match the predictions of an infrared-only model in areas with sun glint.
We first train a pseudo-labeling neural network model using only channels that are unaffected by sun glint. For VIIRS, these channels are M14, M15, and M16. In addition to these VIIRS channels, we also use a subset of the VIIRS/CrIS fusion estimates of MODIS-like channels (MODIS bands 27-36, Table 2) that are similarly unaffected by sun glint, the binary land/water mask and the absolute value of latitude. The VIIRS/CrIS channels are included in an effort to make up for the loss of the shortwave and shortwave infrared VIIRS bands (M1-M13). After training, the pseudo-labeling model is then used to make predictions

for SNPP VIIRS scenes with sun glint of angles of less than 40 degrees over water. For this purpose, we select scenes from the fifteenth day of every month in 2018 (a year included in our training dataset). This is done to ensure even representation of seasons and combinations of sun glint angle and latitude. Of these predictions, roughly one million pseudo-labels are randomly sampled without replacement and added to the original training and validation datasets as if they were obtained from CALIOP. No pseudo-labels are added to the testing dataset. The class probabilities for the pseudo-labeled examples are not required to

be equal to 0 or 1. Instead, they are left unmodified in an effort to promote more reliable class probabilities in pixels affected by sun glint from the final neural network model.

In fig. 2 we qualitatively compare the predictions of the NNCM, we train a naive model on only CALIOP data ignoring the fact that sun glint scenes are not represented in order to better illustrate the purpose of pseudo-labeling. The neural network without pseudo-labels does not include solar zenith angle and sun glint zenith angle since these values for sun glint scenes are outside the range of

230 values for these variables included in CALIOP collocations. The inputs to each model are summarized in Table 3.

In Fig. 2 we qualitatively compare the predictions of the NNCM (that is trained with pseudo-labels) to a neural network model that is not trained with these pseudo-labels. Without pseudo-labeling, the high visible reflectivity causes the neural network model to over predict cloud cover in these regions. Even areas far away from the specular point with only marginal sun glint are significantly impacted. This behavior is not surprising because sun glint is an out-of-domain prediction for the

235 neural network without pseudo-labels. This issue is somewhat remedied by including pseudo-labels in training the NNCM (Fig. 2.d). Qualitatively, the ECM (Fig. 2.f) appears to be the least effected by sun glint and most able to correctly discriminate cloud-free from cloudy in the sun glint region. The MVCM (Fig. 2.e) over predicts cloud cover directly over the specular point, but captures small cloud variability surrounding it. The NNCM makes relatively realistic predictions compared to without pseudo-labeling. However, it does not capture small cloud variability around the specular point to the same degree as the ECM.

The pseudo-labeling model likely has low skill in such conditions due to the lack of visible channels and the low contrast between a low-level fractionally cloudy pixel and the background. There appears to be little disagreement between the cloud masks for the larger, more reflective, and colder cloud features.

To summarize, there are three neural network models trained in this work: (1) the NNCM, (2) a neural network without pseudo-labels, and (3) the pseudo-labeling model. The NNCM is the approach we are proposing and evaluating. The neural

network without pseudo-labels and the pseudo-labeling model are developed in support of the NNCM. The only purpose of the neural network without pseudo-labels is to illustrate the need for pseudo-labeling in Fig. 2. The purpose of the pseudo-labeling model is to provide training labels for the NNCM in sun glint scenes. Only the results from the NNCM are analyzed in Sections 4 and 5. In the following section we describe the details behind how the NNCM is trained.

### 3.2 Neural Network Description and Training Details

We use a simple neural network model that consists of Fully Connected (FC) layers, Leaky Rectified Linear Unit activations (Leaky ReLU), Dropout (Srivastava et al., 2014), and a sigmoid activation as the last layer. The architecture of this model is described in Table 4. All except the last FC layer are followed by Leaky ReLU activation and 2.5% Dropout. Dropout is a neural network regularization technique where a fraction of the units in each layer are randomly ignored and helps prevent

over-fitting. For each VIIRS pixel, a centered 3 pixel by 3 pixel image patch from all 20 inputs is passed to layer group 1 (LG1) of Table 4 and through each layer group successively until the last sigmoid activation is reached. The last sigmoid activation bounds the output of the model between 0 (indicating cloud-free) and 1 (indicating cloudy).

The model in Table 4 is the result of a grid search over a fairly small set of hyperparameters. We tested several configurations by multiplying the number of units in all but the last FC layer by 0.25, 0.5, 1.0, and 2.0. We also tested dropout rates of 0%, 2.5%, 5%, and 10%, and Leaky ReLU vs. ReLU activations. This results in 32 model configurations which are each trained and evaluated three times with different randomly initialized weights. Two configurations with double the number of units in the FC layers reported slightly higher validation accuracies compared to that of Table 4 (a difference of 0.05%). However, we judged that the increase in prediction time was not worth the very small gains in performance. Across all model configurations, Leaky ReLU activation was better than ReLU. Dropout percentages larger than 2.5% only helped when models had a twice the number of units in the FC layers.

Data augmentation is a common method to artificially increase the diversity of examples in the training dataset (Shorten and Khoshgoftaar, 2019). This is often performed by creating plausible alternative views of training examples. Data augmentation methods have been critical in improving performance on widely-used computer vision benchmarks (Zhang et al., 2018, for example). In our case, we are limited by the chosen shape and nature of our input to the kinds of augmentations we can apply to our training dataset. For instance, we cannot reasonably scale, zoom, or translate (all common augmentations applied to images) a 3 pixel by 3 pixel image patch where the center values have special meaning. During training, we apply uniformly random 90 degree rotations (0, 90, 180, 270), horizontal flips, and vertical flips.

$$J = -(y \log \hat{y} + (1 - y) \log (1 - \hat{y})) \tag{1}$$

The neural network is trained to minimize binary cross-entropy, $J$ (Eq. 1), where $y$ is the label and $\hat{y}$ is the predicted probability. All inputs are scaled to have zero mean and unit variance with the means and standard deviations calculated from the training dataset. The Adam optimizer is used with its suggested default parameters (Kingma and Ba, 2015), and we did not notice any substantial changes in the final model when other optimization algorithms were used. The learning rate is initially set to $5 \times 10^{-3}$ with a mini-batch size of 4,098 examples. This value is selected using a learning rate range test (Smith, 2017). After each epoch, the model is evaluated on the validation set. The learning rate is reduced by a factor of 10 when the performance on the validation dataset does not improve for 3 epochs. This continues until a learning rate of $1 \times 10^{-6}$ is reached. Training is stopped once the validation performance does not improve for 5 epochs. Both the final model, and the pseudo-labeling model are trained in the same way with the same set of hyperparameters. Although, since the input size is smaller, the pseudo-labeling model has fewer parameters in the first fully connected layer. Using the same set of hyperparameters is not necessarily ideal since the pseudo-labeling model may have a different set of optimal hyperparameters. We did not perform a separate hyperparameter grid search due to the large computational cost.

The development of the NNCM and the following analysis was performed using the TensorFlow (Abadi et al., 2016), NumPy (Harris et al., 2020), SciPy (Virtanen et al., 2020), and Matplotlib (Hunter, 2007) python libraries.

## 4   Results

### 4.1   Validation with CALIOP

When evaluating classification models many performance metrics need to be viewed in context of the class distribution. Otherwise, quantities such as accuracy (ACC, Eq. 4) and true positive rate (TPR, Eq. 2; equivalent to probability of detection) can be misleading. For example, a trivial binary classification model that predicts only the positive class achieves 0.9 ACC and 1.0 TPR in a dataset with a positive/negative class distribution of 0.9 and 0.1 respectively. Thus, while metrics like ACC and TPR are useful, they must be interpreted within the context of the mean cloud fraction.

We calculate the mean cloud fraction for all VIIRS/CALIOP collocations in our 2019 testing dataset over different surface types for both day and night (Fig. 3). For each instance, a cloud fraction value is reported from CALIOP, the NNCM, the MVCM and the ECM. Daytime cloud fractions include collocations where the solar zenith angle is less than 85 degrees. Land and water surface types are determined from the VIIRS level-1 geolocation data product. The presence of sea ice, snow, and permanent snow (primarily Greenland and Antarctica) is determined from the National Snow and Ice Data Center sea ice index included with the CALIOP Cloud Layer products. The cloud fraction estimates are not necessarily representative of the true cloud fraction over these surface types since they only represent VIIRS/CALIOP collocations for 2019. Instead, we use them to compare the relative tendencies of each cloud mask to generally overestimate or underestimate cloud cover for a given surface type.

The NNCM cloud fractions match closely to that of CALIOP with the exception of an underestimate of 7% over nighttime permanent snow. In all other instances the NNCM reports cloud fractions that are within 3% of CALIOP. The MVCM predicts smaller mean global cloud fraction compared to CALIOP. This seems to be due to a combination of slightly overestimating cloud cover over daytime water, and underestimating cloud cover elsewhere. Of particular note are nighttime snow scenes where MVCM underestimates by 17%, nighttime sea ice where it underestimates by 24%, and areas with permanent snow cover during the night where it underestimates by 30%. The ECM predicts roughly similar values to the NNCM with the exception of overestimating cloud cover during the night over sea ice by 12%.

$$TPR = \frac{TP}{P} \tag{2}$$

$$TNR = \frac{TN}{N} \tag{3}$$

$$ACC = \frac{TP + TN}{P + N} \tag{4}$$

$$BACC = \frac{TPR + TNR}{2} \tag{5}$$

In order to evaluate the performance of each cloud masking model, we calculate the balanced accuracy (BACC; Eq. 5) of all cloud masks across each surface type examined in Fig. 3. BACC is the mean of the true positive rate (TPR; Eq. 2), and the true negative rate (TNR; Eq. 3), where TP is the number of correctly identified clouds, P is the number of clouds, TN is the number correctly identified of cloud-free scenes, and N is the number of cloud-free scenes. The advantage of using BACC over ACC (Eq. 4) is that BACC accounts for class imbalance. One example of class imbalance is daytime sea ice scenes where the mean CALIOP cloud fraction is 76%. A trivial model that predicts 100% cloud fraction would obtain 76% ACC, but only 50% BACC over daytime sea ice.

BACC values are calculated for both the filtered (Table 5) and unfiltered (Table 6) datasets. Table 5 represents the most reliable collocations, but this means that fractionally cloudy scenes, cloud edges, and boundary layer clouds are not well represented. The NNCM reports higher BACC over every surface type examined compared to both the ECM and MVCM for the both the filtered and unfiltered datasets. The most notable improvement from the NNCM occurs over sea ice, snow, and permanent snow during both day and night. McNemar's test (McNemar, 1947) is applied to the NNCM and the best operational model (either ECM or MVCM) for each category in both tables with the null hypothesis that there is no difference in predictive performance between the two models. We reject the null hypothesis with a p-value less than 0.001 in every comparison of the NNCM and the best operational model.

In a few cases, there are instances where one operational model has a higher TPR or TNR value than the NNCM for a particular surface type. We find that that when either the ECM or MVCM has a larger TPR value, it is often at the expense of a very low TNR value (and vice-versa for low TPR and high TNR). One notable example of this is nighttime sea-ice where the ECM has a TPR of 93.3% and a TNR of 36.6% in the analysis of the unfiltered data (Table 6). Another is nighttime permanent snow cover where the MVCM has a TPR of 43.6% and a TNR of 92.2%. The NNCM often has the most similar TPR and TNR values. However, this is not always the case. The largest TPR/TNR disparity for the NNCM is over nighttime water where it has a TPR of 93.6% and a TNR of 79.2%. This is a category where the MVCM has a smaller disparity between TPR and TNR, but still overall lower BACC than the NNCM. Generally when a model has a large disparity between TPR and TNR, that is an indicator of severely over-predicting one of the two classes.

Cloud detection ability relies on many factors including the underlying surface and the characteristics of a given cloud. Clouds with low optical depth may have only a small impact on the top-of-atmosphere radiation observed by the imager. Similarly, clouds that are close to the surface, even if they are optically thick, may be difficult to identify due to low thermal contrast with the surface. We calculate the TPR for all collocations as a function of cloud-top pressure and cloud optical depth as estimated from CALIOP (Fig. 4).

As expected, all cloud masks struggle with the identification of clouds that are optically thin and clouds that are close to the surface. The NNCM has the largest TPR values across all cloud-top pressures and optical depths with a few exceptions. In the unfiltered dataset during the day, the MVCM has the highest TPR values for clouds with tops lower than 850 hPa. For the same cloud-top pressures, the NNCM has the highest TPR in the filtered dataset. This may indicate that the MVCM is better able to discriminate small clouds that are close to the surface. However, when these clouds are removed, the NNCM detects a larger portion of the remaining clouds at all cloud-top pressures. During the night, the MVCM severely underestimates cloud

cover for all cloud-top pressures lower than roughly 350 hPa. This is consistent with the overall lower mean cloud fraction for nighttime scenes reported in Fig. 3. When considering optical depth, the NNCM consistently has a larger TPR for all values during the day and night for the filtered dataset. This is also true for the unfiltered dataset with one exception where it is competitive with the MVCM at optical depths less than 0.2 during the day.

There are some differences between Fig. 4 and Tables 5 and 6 that may seem unintuitive. For example, the ECM has much higher TPR during the night compared to the MVCM for all optical depths and all cloud-top pressures. However, its BACC values for all nighttime collocations is slightly less than that of the MVCM. In this case it is helpful to remember that BACC accounts for both clear and cloudy scenes, and weights each class equally. TPR only accounts for the proportion of clouds correctly identified. The MVCM results in the TPR analysis of Fig. 4 appear to be to due to its tendency to underestimate cloud cover during the night over certain surfaces.

We also investigate the TPR of the three cloud masks as a function of cloud type (Fig. 5). The cloud types are obtained from the 1 km CALIOP Cloud Layers product. Overall, the NNCM reports the highest TPR for most cloud types. One exception is the broken cumulus cloud type in the unfiltered dataset for which the MVCM has the highest TPR. This difference for broken cumulus clouds implies that the NNCM has relatively worse performance in fractionally cloudy scenes compared to the MVCM. While these differences are fairly small, they may be indicative of a much larger difference in skill due to the relative unreliability of the unfiltered collocations. When examining the filtered dataset results for these same clouds, we see that the NNCM has the highest TPR. This suggests that the NNCM and the ECM are only better at detecting broken cumulus when they occupy a substantial horizontal area. When there is considerable fine-scale spatial variability, such as in the unfiltered dataset, these results suggest that the MVCM is the most likely to correctly detect a cloud. Besides the broken cumulus cloud type, the NNCM has the highest TPR for both the filtered and unfiltered collocations. The largest differences are observed when comparing cloud masks for the transparent cloud types. Almost no differences are observed for deep convection which are likely optically thick and have high altitude cloud-tops.

As discussed previously, large TPR values do not necessarily indicate skilful models since they can be obtained by over predicting the positive class. The mean cloud fraction values from Fig. 3 offer some evidence that this is not the case for any of these cloud masks in most scenarios. To add additional context, we plot the receiver operating characteristic (ROC) curves under various geographic and solar illumination conditions (Fig. 6). The ROC curve of each cloud mask depicts the TPR and false positive rate (FPR) over a varying threshold applied to their class probabilities. The NNCM and ECM both natively output cloud probabilities. The MVCM includes a clear-sky confidence estimate which we take the compliment of. An ideal model has a high TPR with very low FPR. A random classifier lies along the diagonal in the middle of a typical ROC plot where TPR is equal to FPR (not shown due to our choice of x and y axis limits).

Figure 6 indicates that the NNCM can obtain higher TPR for any specified FPR in every scenario examined. This is true for both the filtered and unfiltered datasets. This result illustrates that the larger TPR values reported by the NNCM are not strictly due to the larger mean cloud fraction compared the MVCM. In addition to Tables 5 and 6, Fig. 6 implies that most of the improvement by the NNCM comes from the high latitudes during the night, but small improvements can still be observed elsewhere. In every scenario the unfiltered results are worse than those of the filtered datasets. The largest discrepancy between

the filtered and unfiltered datasets occurs in the low-latitudes over the ocean. This is likely due to the prevalence of small

broken cumulus clouds that are mostly removed from the unfiltered dataset.

There are a few situations where the actual TPR and FPR of the models (marked by the colored circles in Fig. 6) are in unintuitive locations on the ROC curve. The ECM's FPR is larger than 40% for nighttime water scenes at the middle and high latitudes (not shown due to our choice of x-axis limits). We expect that this is related to the high mean cloud fraction over these regions measured by CALIOP. Given that the naïve Bayesian models behind the ECM require an initial guess, it is

likely that the ECM is relying heavily on climatology in regions where cloud masking is difficult from infrared observations. Overall, it seems that the locations on the ROC curve of the actual TPR and FPR of the NNCM are related to the mean cloud fraction of the different regions. This is particularly true for nighttime scenes, where statistical models may rely more heavily on the background mean cloud fraction. More cloudy regions such as middle and high latitude nighttime water (with cloud fractions of roughly 79%) have larger FPR. Conversely, nighttime low-latitude land (with a cloud fraction of 50%) has a much

lower FPR. Applications that require specific TPR or FPR from a cloud mask could tune the thresholds applied to the cloud probabilities to reach their desired values indicated by the ROC curves.

Next we examine the performance as a function of geographical region. The mean ACC on the filtered testing dataset is calculated on a 5 by 5 degree grid (Fig. 7). McNemar's test is used to test the differences in model performance between the NNCM and each operational model at every grid point. Only points with significant differences in model performance (p-

values less than 0.001) are shown (Fig. 7.d, Fig. 7.f). Overall, the NNCM appears to be the least sensitive to latitude. Most large differences between the NNCM and the operational models occur over high latitude land. In particular, the NNCM shows large improvement (10-20% difference) over North America, Greenland, Northeastern Asia, and Antarctica over both the MVCM and ECM. Only small improvement (0-10% difference) is observed over the ocean at low and middle latitudes compared to the MVCM. The NNCM shows mixed results compared to the ECM in tropical ocean. A large contribution to the poor performance

of the MVCM in the Arctic and Antarctic is likely due to the severe underestimation of cloud cover observed during the night at high latitudes.

Similarly, we calculate the mean BACC on the same grid in Fig. 8 using the filtered testing dataset. The BACC values are somewhat noisier since areas with extremely high cloud fraction depend largely on the correct identification of a few cloud-free CALIOP profiles. An example of this is over the Southern Ocean, where the ECM has a large disparity between ACC (Fig. 7.e)

and BACC (8.e). A slight tendency to overestimate cloud cover for this region yields very large differences to the NNCM (Fig. 8.f). Besides this example and some areas where the MVCM improves upon the NNCM in the Southern Ocean, the results are largely similar to those of Fig. 7.

All of the previous analyses in this work rely heavily on an individual cloud mask's effective definition of cloud. A difficulty with comparing different clouds masks is that the definition of a cloud is somewhat subjective at low optical depths and perhaps

depends on the particular application. It is plausible that each cloud mask may be more effective at discriminating clouds around a certain optical depth threshold. Thus, a reasonable argument based on the reported global mean cloud fractions in Fig. 3, and the BACC values in Tables 5 and 6, is that the MVCM, due to its lower global mean cloud fraction, may only be sensitive to clouds with slightly larger optical depths compared to the NNCM and ECM.

In order to further probe the differences in these cloud masks, we recalculate BACC after removing clouds below an increasing lower optical depth threshold from our testing dataset (Fig. 9). The aim of this analysis is to understand how the optical depth of a cloud impacts its detectability by each approach, and identify if certain cloud masks perform better if we remove clouds with trivially low optical depths. Even if two cloud masks are developed around slightly different optical depth-based definitions of a cloud, we can reasonably expect their BACC values to converge when clouds with optical depths above both thresholds are removed.

As expected, when optically thin clouds are removed from our testing dataset, the BACC of all the cloud masks is improved. Consistent with Fig. 6, the filtered dataset has higher BACC for all scenarios. The NNCM reports the highest BACC across all land/water, day/night, and latitude combinations examined with a few key exceptions. In low-latitude nighttime water scenes (Fig. 9.j), the ECM has larger BACC for every cloud optical depth threshold in the unfiltered dataset, but more similar values in the filtered dataset. In daytime land scenes at low latitudes (Fig. 9.a), the ECM has larger BACC values above an optical depth threshold of roughly 0.4 for the unfiltered dataset, but has lower BACC values at most optical depths for the filtered dataset. The fact that the NNCM BACC values are still equal to or larger than the other cloud masks for high optical depth clouds in most scenarios suggests the NNCM is overall more skillful in cloud detection regardless of a reasonable optical-depth based definition of a cloud. Because of this, we can infer that improvements in BACC by the NNCM in Tables 5 and 6 are not solely due to discrepancies in the detection of optically-thin clouds.

It may be initially unintuitive why some of the curves in Fig. 9 vary so little with the removal of optically thin clouds. This is partially due to the choice of BACC as our primary performance metric, but it is also representative of the fact that cloud optical depth is not the only variable controlling the detectability of a cloud. Thermal contrast with the surface also plays a significant role. Often, this can be analysed by examining performance of a given cloud mask as a function of both optical depth and cloud-top height. However, this may be misleading where clouds in inversion layers may be warmer than the underlying surface.

To examine the approximate impact of thermal contrast with the surface, we calculate ACC as a function of the difference between the VIIRS M15 measurement (10.8 µm) and the surface temperature obtained from Global Forecasting System (GFS) twelve-hour forecasts made every six hours (Fig. 10). These surface temperatures are matched to VIIRS observations by linearly interpolating in space and time from the preceding and subsequent GFS forecasts. Given the spatial and temporal resolution of the GFS products, these should only be interpreted as very rough estimates of the surface temperature. The differences are calculated after the removal of clouds below two different cloud optical depth thresholds: 0.3, and 3.0. As expected, all cloud masks perform well where the 10.8 µm measurement is significantly colder than the surface. The performance of all models decreases as the VIIRS $10.8 \mu m$ brightness temperatures become more similar to or larger than the surface temperature. Figure 10.b illustrates that even for optically thick clouds, the performance of both operational models is largely dependent on thermal contrast with the surface. The NNCM appears to be more robust to scenes where the 10.8 µm measurement is similar to or warmer than the surface. This is surprising given that the NNCM is not supplied with any information about surface characteristics other than latitude and whether it is viewing a land or water surface.

## 4.2 Uncertainty Assessment

Class probabilities produced by machine learning models are often used to obtain uncertainty estimates. While these values are typically not the same as true uncertainties, they can be useful for interpreting model output. For binary classification models, an approximation for uncertainty can be usually obtained by examining the distance from the decision threshold. These uncertainty estimates are generally unreliable when predictions are made on inputs that are outside the distribution of the original training dataset. With this significant caveat in mind, we calculate the ACC with respect to the cloud probabilities of the NNCM and ECM, as well as the clear-sky confidence from the MVCM (Fig. 11). A model with a cloud probability threshold of 0.5 is perfectly calibrated if its predictions lie along the line where $ACC = \min(\hat{y}, 1 - \hat{y})$ where $\hat{y}$ is the scalar predicted cloud probability. The MVCM appears to follow a different convention with a decision threshold of 0.95 since that is where the minimum accuracy is reached with respect to the MVCM clear-sky confidence.

Overall, the NNCM appears to be the best calibrated with the ACC on the unfiltered collocations closely following the expected values from a perfectly calibrated model. It is slightly over-confident when predicting cloud probabilities for clear-sky cases in the range of 0.1 to 0.4. The ECM appears to be overconfident for the majority of cloud probability values. The assessment of MVCM accuracy as a function of clear-sky confidence is somewhat noisy, but could be attributed to the extremely low number of values in the calculated intervals. Despite the minor differences, all cloud masks examined here have accuracies that vary in an intuitive way with their predicted cloudy or clear-sky probability values. The differences among them can be mostly attributed to how well their class probabilities correspond to a particular level of accuracy. As a result, we expect that these values can be used to convey the relative uncertainty in estimating which imager pixels the CALIOP cloud products might determine to be cloudy. However, it remains to be demonstrated if accurate uncertainties in predicting CALIOP cloud detection translate well to accurate uncertainties outside of CALIOP collocations.

## 4.3 Cloud Detection Consistency

Evidenced by much of the previous analysis, the detectability of a cloudy pixel by a cloud masking algorithm can depend on a number of factors including surface characteristics, solar illumination, cloud optical depth, cloud-top height, thermal contrast with the surface, and the algorithm itself. The variation of BACC, ACC, TPR, and FPR across these conditions suggest that clouds of a fixed optical depth may be more likely detected over certain surface conditions or time of day. This is potentially problematic and conducive to spatial and temporal artifacts in cloud amount analyses. Consider for example, a cloud of fixed low optical depth advected sequentially over a cold land surface, a relatively warm ocean surface, and sea ice. Regardless of the overall accuracy of a cloud mask or effective definition of a cloudy scene, an algorithm with a varying TPR over these surface types could produce spatial artifacts related to these surfaces. Considering that solar illumination may change during this time further complicates this example and could produce unrealistic cloud amount variability through time. In many scenarios, this is unavoidable due to the limitations of the satellite instrument. However, we argue that a desirable quality of a cloud mask is consistency in TPR across varying surface types and solar illumination conditions, and that, ideally, cloud detection should be dependent on characteristics of the cloud and not characteristics of the surface or solar illumination. We expect

that examining TPR differences between these conditions at fixed cloud optical depths could help reveal artificial spatial and temporal variability in cloud amount analyses.

To investigate this concern, we calculate the TPR for clouds above an increasing optical depth threshold. Then, we find the difference in TPR across daytime, nighttime, land, and water for three latitude bands (Fig. 12). An important consideration for Fig. 12 is that a cloud mask can have low accuracy, but also low TPR differences if it makes consistent predictions with respect to cloud optical depth across the conditions examined.

In general, as the lower optical depth threshold increases, TPR differences decrease for all cloud masks with a few exceptions. The NNCM has TPR differences less than or equal to 5% for all scenarios examined except for the difference between nighttime water and nighttime land, and the difference between daytime land and nighttime land at the high latitudes. In both instances, the differences converge to less than 5% at optical depths greater than 1. All cloud masks struggle with consistency at high latitudes and for optically thin clouds.

The ECM shows strong consistency in TPR between daytime and nighttime water at all latitudes for both datasets. However, it struggles in many other scenarios. In Fig. 12.d (low latitude nighttime water and nighttime land), the ECM is the only mask with differences greater than 5%. In Fig. 12.f (high latitude nighttime water – nighttime land) the ECM has the largest TPR difference observed of roughly 28% for optically thin clouds.

The MVCM has the largest TPR differences in nine out of the twelve scenarios examined in Fig. 12. In a few cases (Fig. 12.a, 12.b, 12.g) the large TPR differences converge to zero at larger optical depths. However, in other cases, the large differences remain even for optically thick clouds. This is especially true for daytime/nighttime consistency over both land and water at high latitudes (Fig. 12.i, 12.l) where differences are larger than 10% for clouds with optical depths greater than 1.0.

## 4.4 Regional Analysis

In order to give some context to the largest differences we have observed when validating with CALIOP collocations, we perform a limited regional analysis comparing the NNCM and the MVCM. We focus this analysis on Greenland because it is one of the worst performing regions for both masks. We process every S-NPP VIIRS scene in 2019 where the nadir VIIRS ground track comes within the bounding box of latitudes 60N to 80N and longitudes 70W to 20W. This results in a total of 4,412 six-minute VIIRS scenes. Due to the large amount of scenes, we additionally subsample every fifth pixel from every fifth scanline. For the NNCM and the MVCM we calculate the mean cloud fraction for the region 58N to 84N, and 80W to 10W using a grid size of 0.5 degrees latitude and 1 degree longitude (Fig. 13.a, 13.b).

Consistent with the CALIOP validation, we observe large differences over the Greenland land mass (Fig. 13.c). The NNCM predicts 10-25% higher cloud fraction over Greenland varying with exact location. Differences over the ocean to the southeast of Greenland are negative and fairly small. However, the ocean to the north and west of Greenland have large positive differences similar to those over Greenland itself. Based on the spatial characteristics of the mean MVCM cloud fraction over the ocean, we hypothesize that these differences may be a result of sea ice cover. A similar result was found previously in Liu et al. (2010), where MODIS cloud detection errors related to the presence of sea ice were suggested to contribute to large errors in cloud fraction trends.

Focused regional comparisons between imagers and CALIOP can be difficult due to the relative sparsity of CALIOP ob-
servations in small geographical regions. A domain-wide averaged cloud fraction comparison between the two imager cloud
masks and CALIOP is subject to a large amount of error due to the differences in spatial sampling and observation times. We
calculate a domain-wide average of cloud fraction for CALIOP and the two cloud masks and plot the 31-day moving average
as a function of time (Fig. 13.d). To account for some of the differences in sampling, this average only includes grid points from
the NNCM and MVCM for which CALIOP has sampled on the same day. This effectively removes the impact of differences
in spatial sampling, but ignores differences in temporal sampling. Thus, we should still not expect either the MVCM or the
NNCM to follow the CALIOP 1 km or 5 km products closely. When calculating the mean cloud fraction, individual values on
the regular latitude/longitude grid are weighted to account for differences in surface area between locations.

        The largest differences occur in northern hemisphere winter, with better agreement between the MVCM and NNCM oc-
curring during northern hemisphere summer. This suggests that the MVCM's tendency to underestimate cloud cover during
conditions with no solar illumination heavily contributes to the spatial differences observed in Fig. 13.c. Similarly, the magni-
tude of the seasonal cycle in the MVCM is likely exaggerated due to variation of solar zenith angle throughout the year. Both
cloud masks also show very different shapes to the seasonal cycle even when ignoring the overall differences in mean cloud
fraction. Despite differences in temporal sampling, the NNCM shows somewhat similar variability to both CALIOP products.
Overall, the NNCM shows mean cloud fractions more similar to the 5 km CALIOP product despite being trained with labels
from the 1 km product. This is not a surprising result since the NNCM is a statistical algorithm and is incentivized to predict
the majority class (cloudy) in uncertain conditions when both classes are given equal weight. The 5 km CALIOP product likely
has a larger mean cloud fraction due to its ability to detect clouds with low optical depths. Of the two cloud masks, the NNCM
appears to give a more realistic assessment of cloud cover variability in this analysis and more closely aligns with that of
CALIOP.

## 5   Discussion

There are few common themes in much of the analysis done in section 4. The BACC calculated over global averages of a few
surface types suggests that the NNCM is better at discriminating cloudy from cloud-free scenes in most scenarios. Further
analysis shows that a large majority of this improvement comes from collocations located at the middle and high latitudes.
According to the CALIOP collocations, the ECM and NNCM cloud masks appear relatively comparable over low-latitude land
and ocean with the MVCM trailing slightly behind both in this region. The ECM appears slightly more capable of identifying
low-level small clouds in the unfiltered dataset in low-latitude nighttime scenes over water. The NNCM's improvement at higher
latitudes raises some questions on its dependence on latitude particularly since it is the only model that uses this information
directly in its inputs. To test this dependency, we retrained and evaluated the NNCM after removing latitude, solar zenith angle,
sun glint angle, and the land/water mask. The largest change in BACC was a decrease of -0.5% over nighttime water, and all
other surfaces changed by less than 0.2%. Considering these results, it is probable that the NNCM depends on latitudinal mean
cloudiness in some capacity over water (similar to the ECM over the Southern Ocean). However, it is difficult to assess how

this information is utilized and whether it is serving a purpose similar to that of a climatological first guess, or if it is changing the usage of other observations.

Despite training using an unfiltered dataset that contains fractionally cloudy pixels identified by CALIOP, the NNCM still struggles in fractionally cloudy scenes. This is likely due to a combination of noisy labels from CALIOP in these conditions and the low contrast with the underlying and surrounding surface. Broken cloudiness is a consistent problem in using CALIOP as a reference. These clouds pose a significant challenge to cloud masking in general, but are particularly difficult to handle when the corresponding CALIOP profile is not fully representative of its collocated imager pixel. Future efforts to provide a high-quality, fine-resolution, globally-distributed cloud labels could prove extremely useful to solve these issues. Our choice of training on an unfiltered collocation dataset was made to avoid any bias with regards to the spatial characteristics of cloud cover. We expect that filtering out spatially variable clouds from the training dataset would result in an even worse characterization of small clouds by the NNCM. Despite training on a relatively unreliable collection of CALIOP collocations, we report much higher BACC for the vast majority of scenarios, especially in homogeneously cloudy scenes represented by the filtered testing dataset.

It should also be noted that the decision to use CALIOP as a reference and the lack of filtering applied to the training dataset affects how the NNCM uncertainty estimates can be interpreted. Reported uncertainties by the NNCM should not be purely attributed to the ability of the model to detect clouds based on spectral variability alone. Since we include neighboring pixels in the inputs, spatial variation in VIIRS channels is also a contributor. Additionally, these uncertainty estimates are also a function of how representative CALIOP profiles typically are of a given pixel. This suggests that uncertainties associated with regions of broken clouds are elevated due to the difficulty of obtaining mutually representative collocations between CALIOP and VIIRS.

There are many areas for improvement in the NNCM approach. For instance, we included all 16 moderate resolution channels in our algorithm. It is plausible that some channels are not especially useful in cloud detection, or the useful information they provide to the task is redundant among other channels. Pruning inputs to the model could ultimately speed up processing and could reduce the likelihood of over-fitting. Future work could investigate the benefit of including the 375 m I-band measurements from VIIRS. We did not include I-band measurements, since obtaining these observations more than doubled the processing time for creating the collocation dataset, training the model, and making predictions. Sub-pixel information from the I-band measurements could likely help identify small cloud features. However, we expect that the poor representation of small clouds by the CALIOP/VIIRS collocations would severely limit the usefulness of their incorporation. Further work is needed to in order to properly assess how I-band measurements could be used to maximize their value in cloud property algorithms trained with CALIOP.

Despite the large increase in BACC made by our NNCM approach, there is still room for improvement particularly during the night. One potential solution might be the incorporation of VIIRS/CrIS fusion channels into the inputs of the final NNCM model. Similar to the usage of I-band measurements, this may increase the prediction time. However, the spectral regions covered by the I-bands are already well-represented in the moderate resolution channels. The VIIRS/CrIS fusion channels represent spectral regions not covered in the native VIIRS channels such as those with significant $CO_2$ (MODIS bands 33-36) , $H_2O$ (MODIS bands 27 and 28), and $O_3$ (MODIS band 30) absorption. Thus, the increase in cloud detection accuracy may be

worth the trade-off of increased prediction time associated with their inclusion. However, an added difficulty is that the fusion channel estimates are made from relatively coarse resolution CrIS channels. This could negatively impact cloud detection for

fractionally cloudy pixels to an even greater degree.

Our approach currently includes very little ancillary data: only a VIIRS-derived binary land/water mask. The MVCM uses several, including surface temperatures, sea ice, snow cover, and Normalized Difference Vegetation Index maps. The ECM also includes surface temperatures, sea ice, snow cover, tropopause temperatures, and clear-sky estimates of many channels using radiative transfer models. Anecdotally, we notice that some spatial artifacts we have observed in the two operational

cloud masks appear to be related to the relatively coarse resolution of the ancillary datasets. Early experiments with the neural network lead us to believe that including surface temperature increased the frequency of spatial artifacts in its output. This motivated our decision to initially not include information such as surface temperatures in our approach even though it lead to substantial increases in cloud detection performance estimated by CALIOP collocations. The relatively coarse-resolution of the ancillary data might cause issues around boundaries of surface types or around large horizontal gradients in surface

temperature. This mischaracterization of the surface condition could result in errors in cloud detection if a given model is highly dependent on this information. This is potentially one of the explanations for the disparity in performance in instances of low thermal contrast with the surface. We leave it to future work to investigate how to include coarse-resolution ancillary data in the neural network without increasing the prevalence of spatial artifacts in cloud masking output.

For all scenarios examined in Fig. 12 we conclude that the NNCM is the most consistent in identifying clouds across various

geographical, solar illumination, and surface conditions while controlling for cloud optical depth. There are several reasons why the NNCM model might be successful in this regard. The ECM and MVCM both apply different tests based on surface condition and solar zenith angle. The ECM, for example, is a collection of naïve Bayesian models trained for different surface types. This a very intuitive approach, but in practice requires partitioning collocation datasets according to surface type and reduces the number of collocations that can be used for training each model. Similarly the MVCM uses different decision

pathways and restricts or requires usage of certain inputs accordingly. We hypothesize that training a only single model (rather than multiple), and instead providing the land/water mask and solar-zenith angle as inputs has contributed to its consistency in cloud detection under these varying conditions.

In one of the worst performing regions for all cloud masks, we observe very substantial differences in mean cloud fraction for 2019 across both space and time. These results demonstrate how differences in TPR of a cloud mask over varying surface

and illumination conditions could potentially contribute to very different spatial and temporal variability. Because of this, we argue that TPR differences over varying surface and illumination conditions could be useful metrics for identifying such issues in cloud mask development and assessment. We suspect that this is a particularly important consideration for the use of cloud masking approaches in climate records. For example, annual sea ice loss or trends in seasonal snow cover could produce erroneous trends in cloud cover if a given cloud mask's TPR differs significantly to that of ice-free ocean or snow-free land.

We note several potential caveats in the assessment of the NNCM in addition to issues with fractional cloudiness. One clear limitation with using CALIOP as a source for labels is the relatively narrow range of sensor viewing angle and solar illumination combinations. We examined one specific example of this in sun glint and have limited, but not completely removed, its adverse

impact on cloud detection using pseudo-labeling. One disadvantage of the pseudo-labeling approach, is that the associated uncertainty estimates lose much of their meaning in domains where we exclusively train on pseudo-labels. We have attempted to limit the impact of this issue by training the NNCM to estimate the class probabilities produced by the pseudo-labeling model, and not the predicted class labels themselves. This approach appears to be successful in preventing severe over-clouding of sun glint regions, but it can only be expected to perform as well as a model that uses infrared observations exclusively. There are very specific conditions in which the two operational masks outperform the NNCM and it may be possible to use MVCM or ECM predictions as pseudo-labels to address deficiencies in the NNCM if these conditions can be identified without the use of CALIOP. We have not evaluated how the NNCM performs specifically in cloud-free scenes with high aerosol loading in this analysis. We expect that the ability for CALIOP to distinguish cloud from aerosol layers could add an another layer of difficulty in addition to the ability of VIIRS observations to distinguish these features.

One source of bias in this assessment is our choice of using the 1 km CALIOP Cloud Layers products in the vast majority of our comparisons. It is possible that some optically thin clouds that are detected in the 5 km CALIOP product but are missed in the 1 km product could be correctly identified by the imager cloud masks. This is plausible in conditions such as daytime low-latitude ocean where a thin cirrus cloud has large thermal contrast with the surface. We have not investigated this specific concern in this work due to the difficulty of ensuring mutually representative collocations between the 5 km CALIOP product and the 750 m observations. It is possible that the slight overestimation in daytime mean cloud fraction by the MVCM (Fig. reffig:fig03) could be due to the detection of clouds missed by the 1 km CALIOP product. For purely statistical approaches, like the NNCM, it is difficult to separate this possibility from that of over-predicting cloud fraction simply because cloudy scenes are more common than cloud-free.

## 6 Conclusions

In this work, we examine the performance of a neural network cloud mask (NNCM) for VIIRS that is trained with coincident CALIOP observations and compared it with two operational cloud masks. Both the MVCM and ECM appear to be slightly better at identifying small broken clouds than the NNCM. However, the NNCM outperforms both operational masks in most other conditions. We observe particularly large improvement at the middle and high latitudes during the night where the operational masks missed substantial fractions of optically-thick clouds that were correctly identified by the NNCM. We have ruled out the possibility that the improvement is due to disagreements in each approach's effective definition of a cloud. Furthermore, we find that uncertainty estimates from the NNCM are well-calibrated and appropriately represent the ability to estimate cloudy or cloud-free labels from CALIOP. When examining differences in true positive rate, we find that the NNCM is the most consistent in identifying clouds of a fixed optical depth when considering day/night and land/water conditions. A regional analysis over Greenland for 2019 confirms that such differences could contribute to vastly different assessments of the spatial and temporal variability of cloud cover over certain regions. Some issues with the global representativeness of VIIRS/CALIOP collocations are successfully mitigated with a simple semi-supervised learning approach, but more work is needed in improving detection of fractionally cloudy pixels by the NNCM.

*Author contributions.* C.W. developed the neural network, completed the analysis, and wrote the manuscript. A.H. and S.A. supervised the project, and contributed to the review and editing of the paper. All authors contributed to the initial conceptualization and planning of this work.

*Competing interests.* The authors declare that they have no conflict of interest.

*Acknowledgements.* The authors would like to acknowledge Denis Botambekov's help in running the Enterprise Cloud Mask. Additionally we wish to thank the UW Madison SSEC/CIMSS Atmosphere SIPS team sponsored under NASA contracts NNG15HZ38C and 80GSFC20C097 for providing software tools for easily obtaining VIIRS data. This work was supported by the JPSS program under NOAA CA NA15NES4320001. The views, opinions, and findings contained in this report are those of the authors and should not be construed as an official National Oceanic and Atmospheric Administration or U.S. Government position, policy, or decision.

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

| Band | Spectral Range ($\mu$m) | Units |
|------|-------------------------|-------|
| M1   | 0.400 - 0.421           | Refl. |
| M2   | 0.436 - 0.451           | Refl. |
| M3   | 0.477 - 0.496           | Refl. |
| M4   | 0.541 - 0.561           | Refl. |
| M5   | 0.662 - 0.680           | Refl. |
| M6   | 0.738 - 0.752           | Refl. |
| M7   | 0.843 - 0.881           | Refl. |
| M8   | 1.225 - 1.252           | Refl. |
| M9   | 1.368 - 1.383           | Refl. |
| M10  | 1.571 - 1.631           | Refl. |
| M11  | 2.234 - 2.280           | Refl. |
| M12  | 3.598 - 3.791           | BT [K] |
| M13  | 3.987 - 4.145           | BT [K] |
| M14  | 8.407 - 8.748           | BT [K] |
| M15  | 10.234 - 11.248         | BT [K] |
| M16  | 11.405 - 12.322         | BT [K] |

**Table 1.** The band, spectral range, and units of all sixteen moderate resolution VIIRS channels. Each channel is expressed as a reflectivity (Refl.), or a brightness temperature (BT).

| VIIRS/CrIS Fusion Channel | Spectral Range of MODIS Equivalent Channel ($\mu$m) |
|---|---|
| MODIS 27 | 6.535 – 6.895 |
| MODIS 28 | 7.175 – 7.475 |
| MODIS 29 | 8.400 – 8.700 |
| MODIS 30 | 9.580 – 9.880 |
| MODIS 31 | 10.780 – 11.280 |
| MODIS 32 | 11.770 – 12.270 |
| MODIS 33 | 13.185 – 13.485 |
| MODIS 34 | 13.485 – 13.785 |
| MODIS 35 | 13.785 – 14.085 |
| MODIS 36 | 14.085 – 14.385 |

**Table 2.** The VIIRS/CrIS fusion channels used in the pseudo-labeling model. All channels are expressed as brightness temperatures.

| Inputs | NNCM | Neural network without pseudo-labels | Pseudo-labeling model |
|---|---|---|---|
| M1-M13 | X | X | |
| M14-M16 | X | X | X |
| MODIS 27 - MODIS 36 | | | X |
| \| Latitude \| | X | X | X |
| Solar Zenith Angle | X | | |
| Sun Glint Angle | X | | |
| Land/Water Mask | X | X | X |

**Table 3.** Summary of the inputs included in the three neural networks used in this work. See the main text for description of each model.

| Layer Group (LG) | Layer Type | Input Size | Output Size |
|---|---|---|---|
| LG1 | FC(200), Leaky ReLU, Dropout(2.5%) | 180 (3x3x20) | 200 |
| LG2 | FC(200), Leaky ReLU, Dropout(2.5%) | 200 | 200 |
| LG3 | FC(100), Leaky ReLU, Dropout(2.5%) | 200 | 100 |
| LG4 | FC(50), Leaky ReLU, Dropout(2.5%) | 100 | 50 |
| LG5 | FC(25), Leaky ReLU, Dropout(2.5%) | 50 | 25 |
| LG6 | FC(1), Sigmoid | 25 | 1 |

**Table 4.** The architecture of the NNCM. LG refers to Layer Group and is used to describe the collection of layers in each row. FC(x) refers to the fully connected layers where x is the number of units in each layer. Similarly, Dropout(x) refers to the fraction of inputs which dropout is applied.

| | NNCM | | | ECM | | | MVCM | | | Cloud | Number |
|---|---|---|---|---|---|---|---|---|---|---|---|
| | BACC | TPR | TNR | BACC | TPR | TNR | BACC | TPR | TNR | Fraction | (Million) |
| Day Global | 0.968 | 0.982 | 0.954 | 0.938 | 0.957 | 0.918 | 0.910 | 0.941 | 0.879 | 0.662 | 2.96 |
| Night Global | 0.934 | 0.960 | 0.908 | 0.849 | 0.927 | 0.772 | 0.876 | 0.853 | 0.900 | 0.721 | 2.91 |
| Day Water | 0.969 | 0.985 | 0.952 | 0.940 | 0.977 | 0.902 | 0.909 | 0.966 | 0.852 | 0.735 | 1.99 |
| Night Water | 0.932 | 0.976 | 0.888 | 0.842 | 0.969 | 0.715 | 0.893 | 0.899 | 0.887 | 0.803 | 1.99 |
| Day Land | 0.965 | 0.974 | 0.956 | 0.917 | 0.898 | 0.936 | 0.887 | 0.866 | 0.908 | 0.512 | 0.97 |
| Night Land | 0.916 | 0.906 | 0.927 | 0.808 | 0.791 | 0.825 | 0.808 | 0.705 | 0.912 | 0.542 | 0.91 |
| Day Sea Ice | 0.966 | 0.966 | 0.966 | 0.883 | 0.962 | 0.804 | 0.879 | 0.859 | 0.899 | 0.775 | 0.29 |
| Night Sea Ice | 0.895 | 0.932 | 0.859 | 0.661 | 0.944 | 0.379 | 0.790 | 0.663 | 0.917 | 0.757 | 0.31 |
| Day Permanent Snow | 0.961 | 0.964 | 0.959 | 0.885 | 0.840 | 0.929 | 0.822 | 0.739 | 0.905 | 0.421 | 0.30 |
| Night Permanent Snow | 0.863 | 0.832 | 0.895 | 0.701 | 0.671 | 0.731 | 0.694 | 0.461 | 0.927 | 0.578 | 0.36 |
| Day Snow Land | 0.954 | 0.961 | 0.947 | 0.855 | 0.859 | 0.852 | 0.864 | 0.825 | 0.903 | 0.631 | 0.16 |
| Night Snow Land | 0.920 | 0.927 | 0.913 | 0.758 | 0.827 | 0.688 | 0.778 | 0.675 | 0.880 | 0.617 | 0.19 |

**Table 5.** BACC, TPR, and TNR calculated for each cloud mask over different surfaces during day and night for the filtered dataset. Collocation counts do not sum to the count listed in the "All" row because sea ice collocations are also counted in the water category, and the two snow categories are also counted in the land category. Cloud fraction is calculated from the CALIOP collocations.

| | NNCM | | | ECM | | | MVCM | | | Cloud | Number |
|---|---|---|---|---|---|---|---|---|---|---|---|
| | BACC | TPR | TNR | BACC | TPR | TNR | BACC | TPR | TNR | Fraction | (Million) |
| Day Global | 0.905 | 0.934 | 0.877 | 0.879 | 0.906 | 0.853 | 0.851 | 0.902 | 0.801 | 0.635 | 3.63 |
| Night Global | 0.879 | 0.920 | 0.838 | 0.808 | 0.889 | 0.726 | 0.830 | 0.816 | 0.843 | 0.687 | 3.46 |
| Day Water | 0.900 | 0.937 | 0.863 | 0.876 | 0.930 | 0.822 | 0.842 | 0.935 | 0.749 | 0.691 | 2.48 |
| Night Water | 0.864 | 0.936 | 0.792 | 0.796 | 0.930 | 0.663 | 0.832 | 0.860 | 0.804 | 0.747 | 2.45 |
| Day Land | 0.910 | 0.925 | 0.895 | 0.865 | 0.835 | 0.895 | 0.839 | 0.807 | 0.871 | 0.515 | 1.16 |
| Night Land | 0.884 | 0.870 | 0.899 | 0.782 | 0.754 | 0.810 | 0.783 | 0.671 | 0.895 | 0.542 | 1.01 |
| Day Sea Ice | 0.931 | 0.941 | 0.922 | 0.851 | 0.944 | 0.759 | 0.852 | 0.832 | 0.872 | 0.757 | 0.31 |
| Night Sea Ice | 0.870 | 0.906 | 0.834 | 0.650 | 0.933 | 0.366 | 0.772 | 0.640 | 0.903 | 0.741 | 0.33 |
| Day Permanent Snow | 0.930 | 0.928 | 0.932 | 0.854 | 0.790 | 0.917 | 0.795 | 0.692 | 0.899 | 0.430 | 0.32 |
| Night Permanent Snow | 0.836 | 0.797 | 0.875 | 0.684 | 0.646 | 0.722 | 0.679 | 0.436 | 0.922 | 0.577 | 0.40 |
| Day Snow Land | 0.905 | 0.920 | 0.891 | 0.818 | 0.815 | 0.820 | 0.827 | 0.779 | 0.875 | 0.619 | 0.19 |
| Night Snow Land | 0.887 | 0.890 | 0.885 | 0.737 | 0.797 | 0.678 | 0.756 | 0.641 | 0.870 | 0.610 | 0.21 |

**Table 6.** Same as Table 5, but all metrics are computed for the unfiltered collocations.

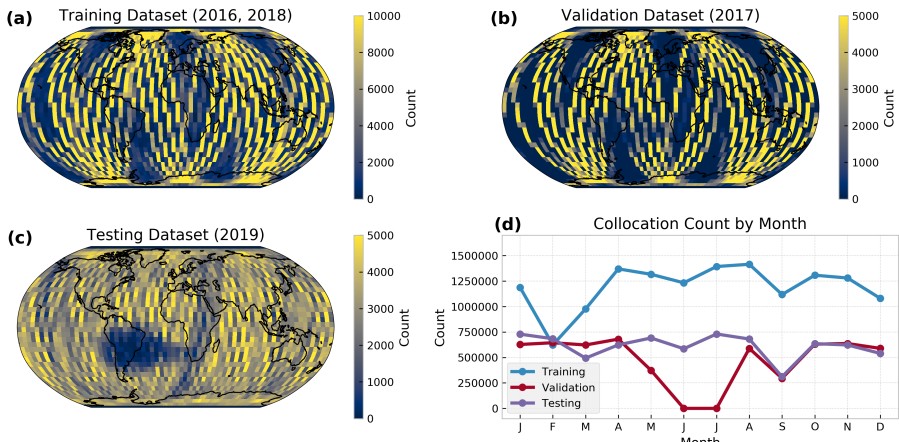

**Figure 1.** Spatial distribution of the unfiltered S-NPP VIIRS/CALIOP collocations for the (a) training, (b) validation, and (c) testing datasets. Panel (d) indicates the seasonal distribution of collocations for each unfiltered dataset. Note the difference in color bar limits between (a), (b), and (c).

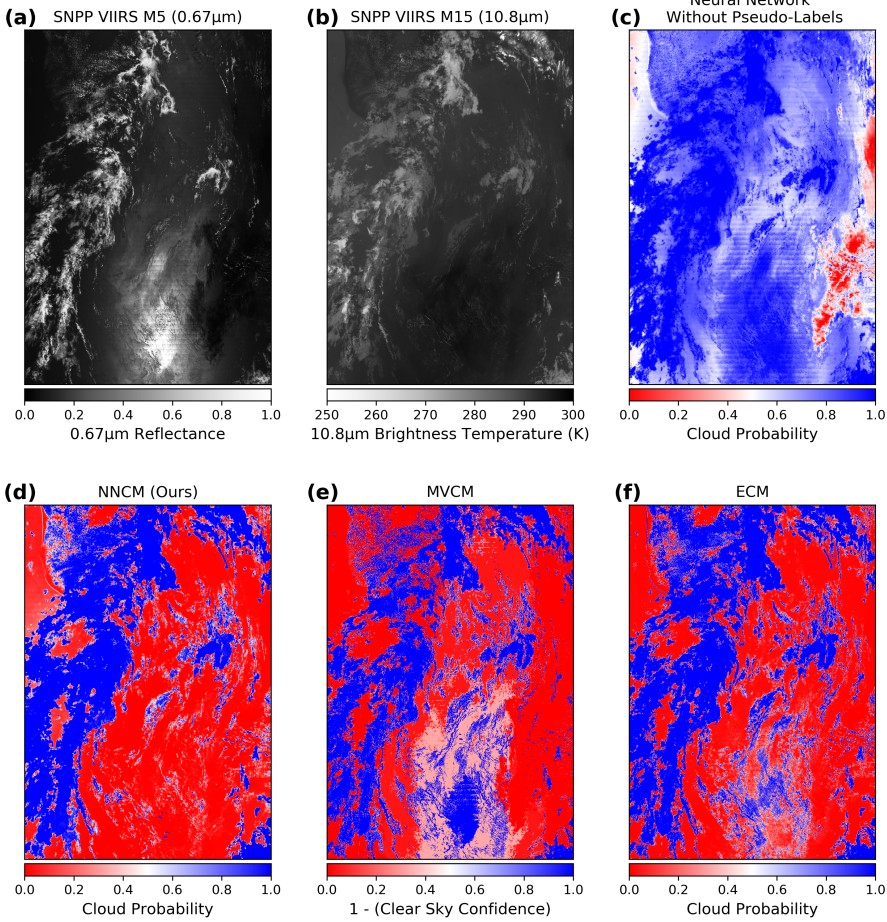

**Figure 2.** Comparison of the neural network cloud mask without pseudo-labels (c),the NNCM (d), the MVCM (e), and the ECM (f). Also shown are band M5 with a central wavelength of roughly $0.67\mu$m (a) and band M15 with a central wavelength of roughly $10.8\mu$m (b).

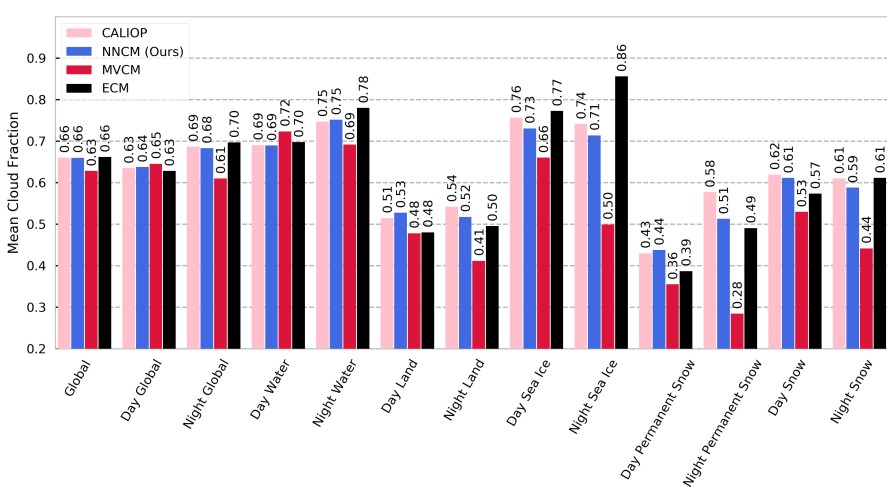

**Figure 3.** Mean cloud fraction for the 2019 unfiltered testing dataset. Each bar grouping from left to right shows the value from the CALIOP 1 km product, the NNCM, MVCM, and ECM. Time of day and surface categorizations are described in the main text.

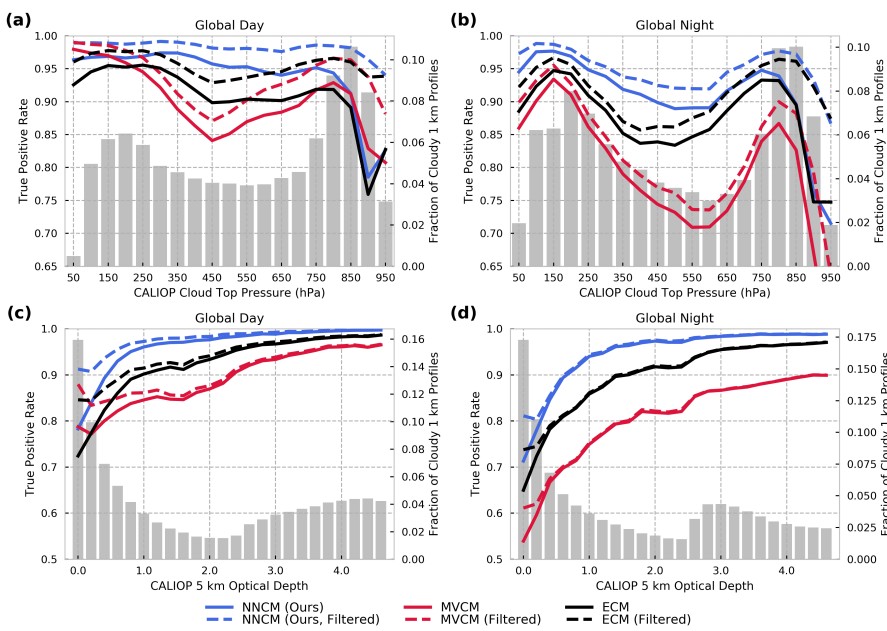

**Figure 4.** True positive rate (TPR) calculated as function of cloud-top pressure (a,b) and optical depth (c,d) for daytime and nighttime collocations respectively. The grey bars represent the fraction of cloudy 1 km CALIOP profiles. Only profiles with non-zero optical depths are included in (c) and (d).

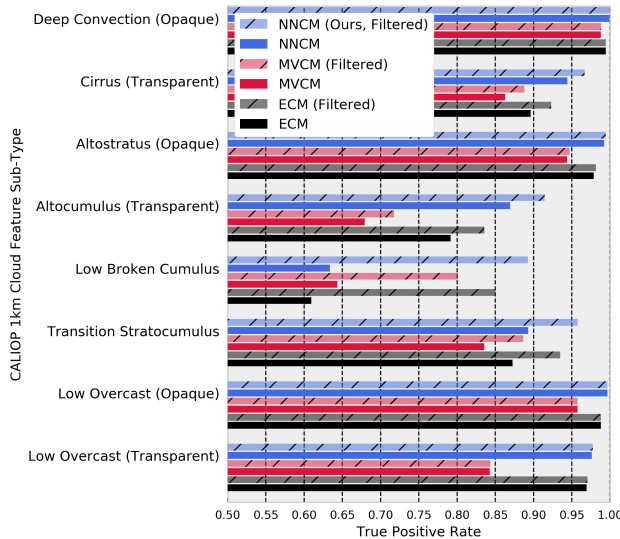

**Figure 5.** The True Positive Rate (TPR) for various CALIOP cloud-feature types from the 1 km CALIOP Cloud Layers product. The order shown in the legend indicates the ordering of the bars in each grouping.

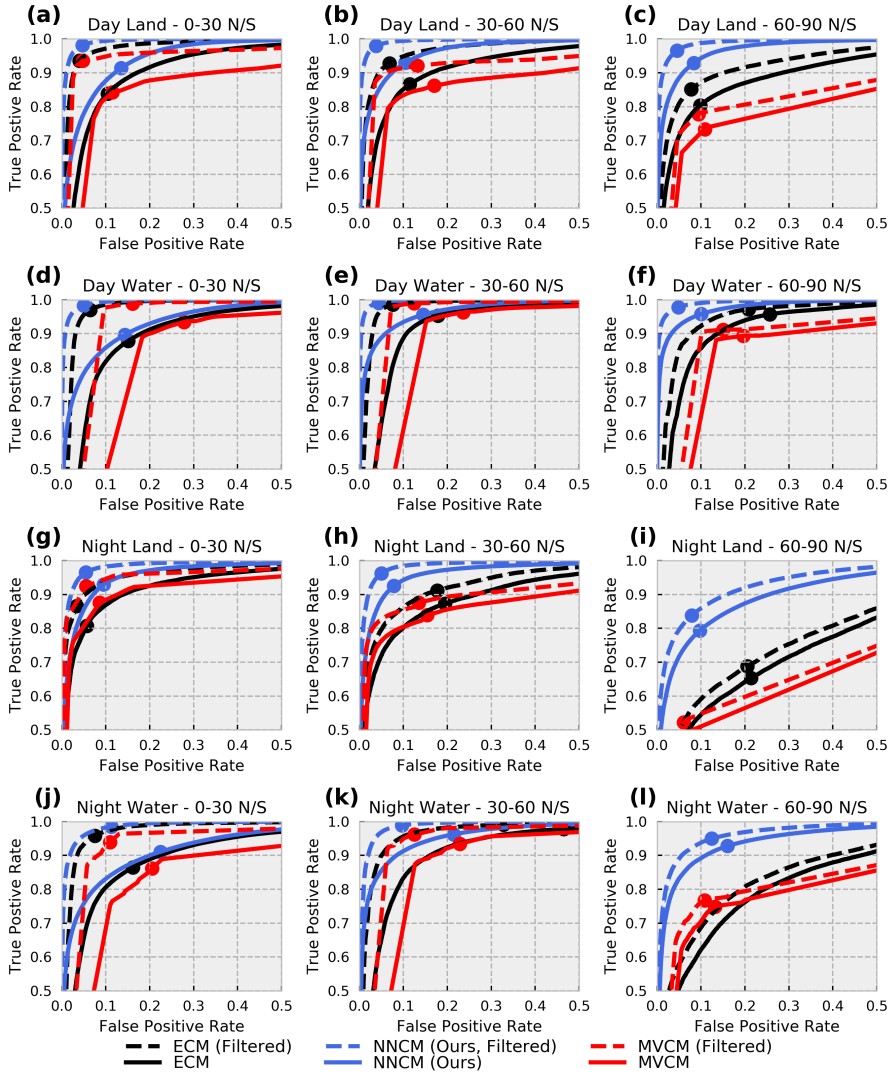

**Figure 6.** Receiver operating characteristic (ROC) curves for all three cloud masks. The text above each subplot indicates the subset of collocations for which the curves are plotted. Note that the x and y axis limits are somewhat atypical for ROC curve plots and are chosen here to emphasize the differences between the masks and different datasets. The TPR and FPR for the model using the standard threshold of 0.5 for the neural network and ECM, as well as the integer cloud mask for MVCM are also shown with similarly colored circles.

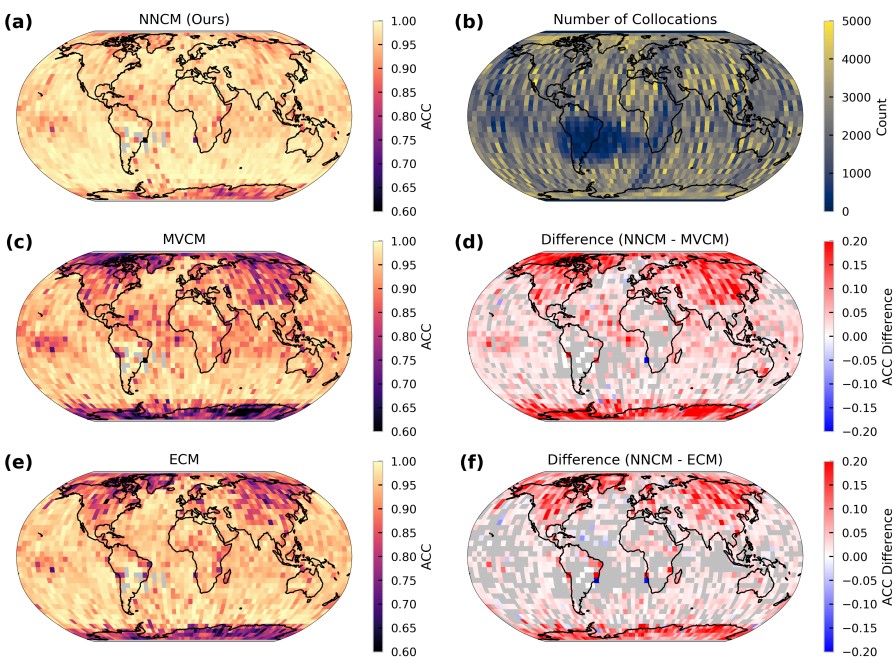

**Figure 7.** Geographic comparison of the ACC between the three cloud masks on the filtered testing dataset. Each grid cell is 5 degrees latitude by 5 degrees longitude. The gap in coverage over South America is due to the removal of low-energy laser shots from the CALIOP datasets. Cells with less than 100 collocations are not shown in (a) or (c)-(f). Differences are only shown where determined significant by McNemar's test with p-values less than 0.001.

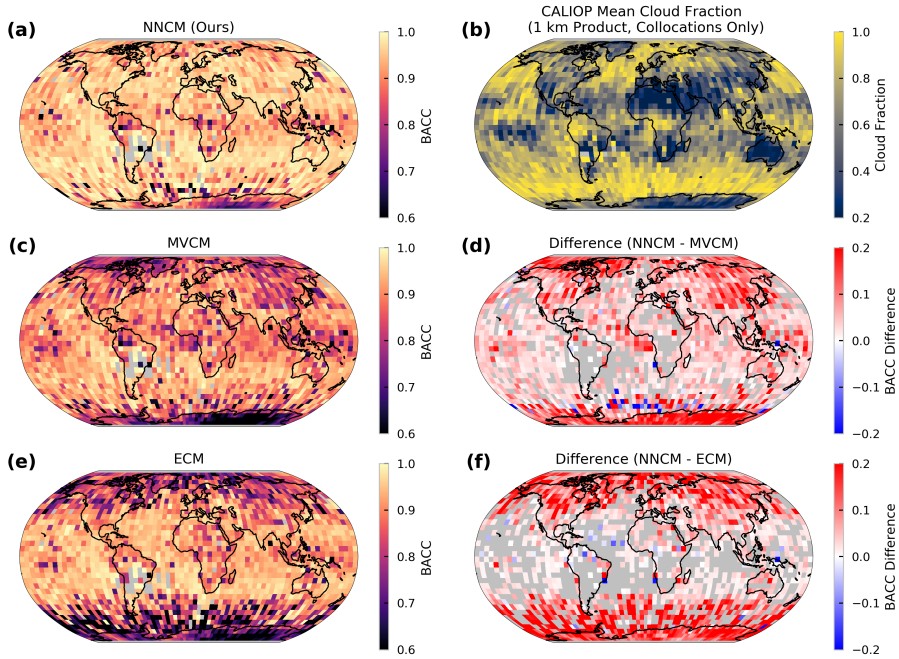

**Figure 8.** Same as Fig. 7 but all using BACC instead of ACC. Panel (b) has been replaced with the 1 km CALIOP cloud fraction computed from the VIIRS/CALIOP collocations.

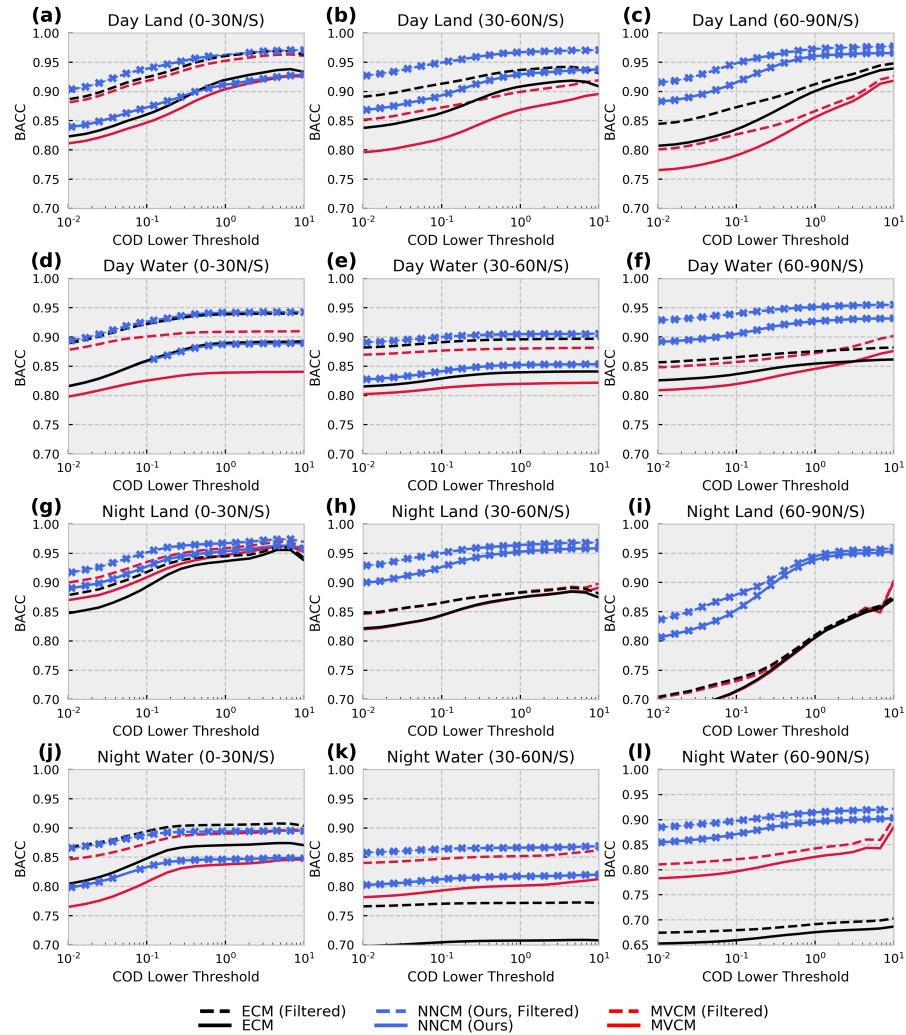

**Figure 9.** Balanced Accuracy (BACC) recalculated after removing clouds below a certain cloud optical depth (COD) threshold. Tick marks on the neural network lines indicate significant differences in performance between the neural network and the best operational model using McNemar's test with p-values less than 0.001. Note that the y-axis limits are different for (l) compared to the other subplots.

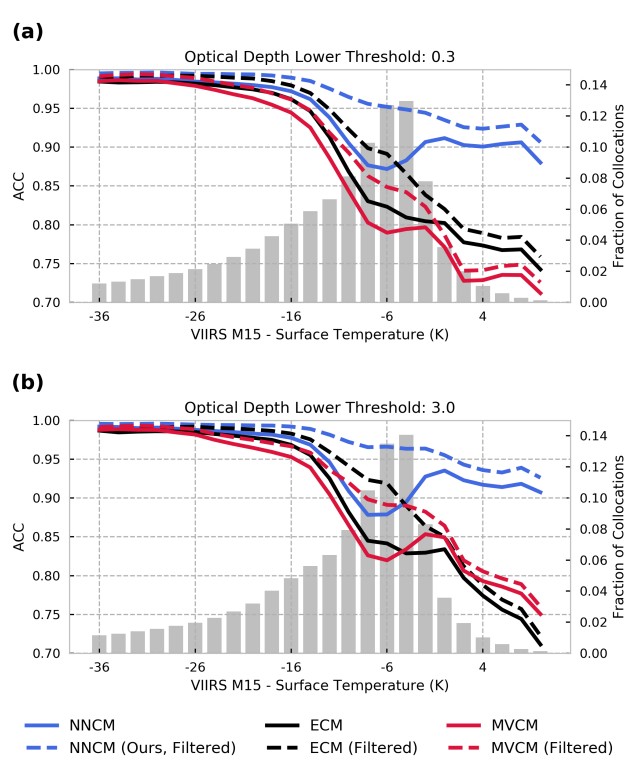

**Figure 10.** ACC calculated as a function of thermal contrast with the surface approximated by the difference between VIIRS M15 (10.8 μm) and surface temperature in Kelvin. Each subplot represents a set of collocations consisting of clear-sky scenes and cloudy scenes with optical depths greater than 0.3 (a) and 3.0 (b).

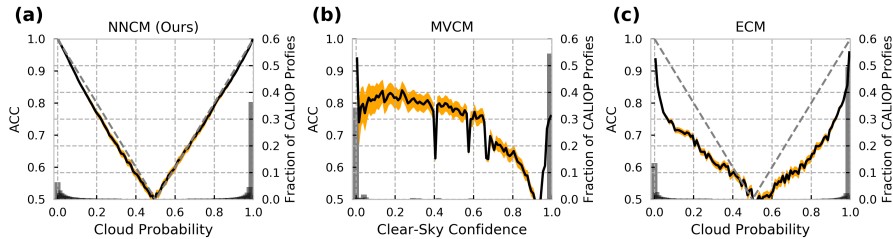

**Figure 11.** Uncertainty assessments for (a) the NNCM (b) the MVCM, and (c) the ECM. ACC values (left y-axis) for cloud probability and clear sky confidence values are calculated for bins of size 0.01. For (a) and (c) a perfectly-calibrated model is plotted with the grey dashed line (see main text). Orange shading indicates the 99.9% confidence interval. Grey bars indicate the fraction of collocations falling within each bin of width 0.01.

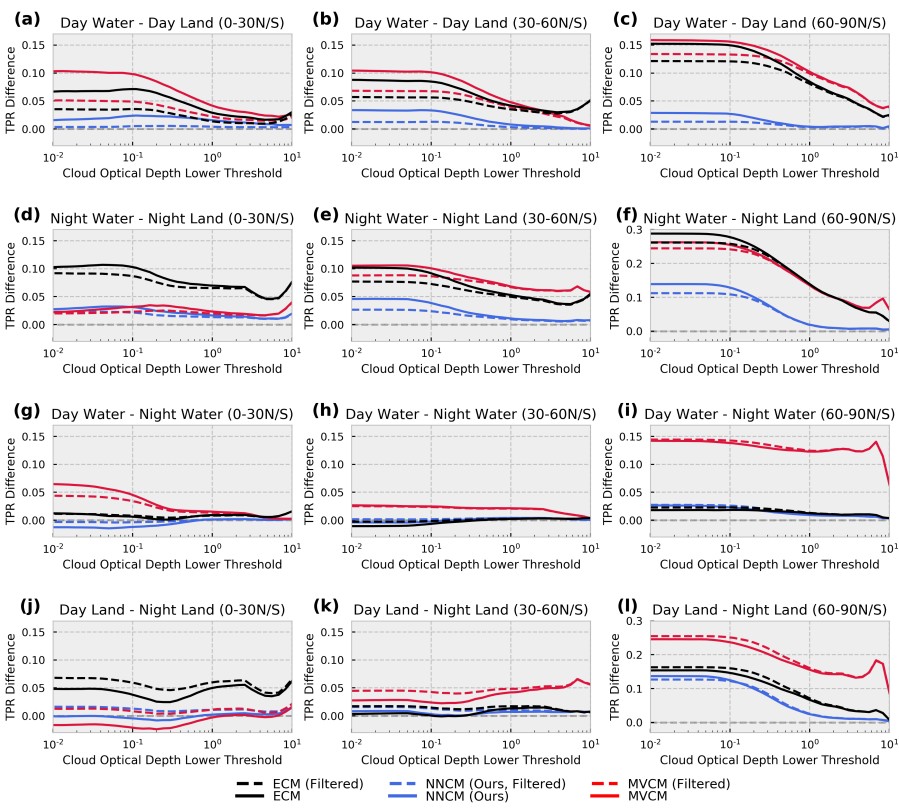

**Figure 12.** TPR differences over combinations of land/water and day/night conditions. The specific TPR difference and latitude is labeled at the top of each subplot. Note that the y-axis limits are different for (f) and (l).

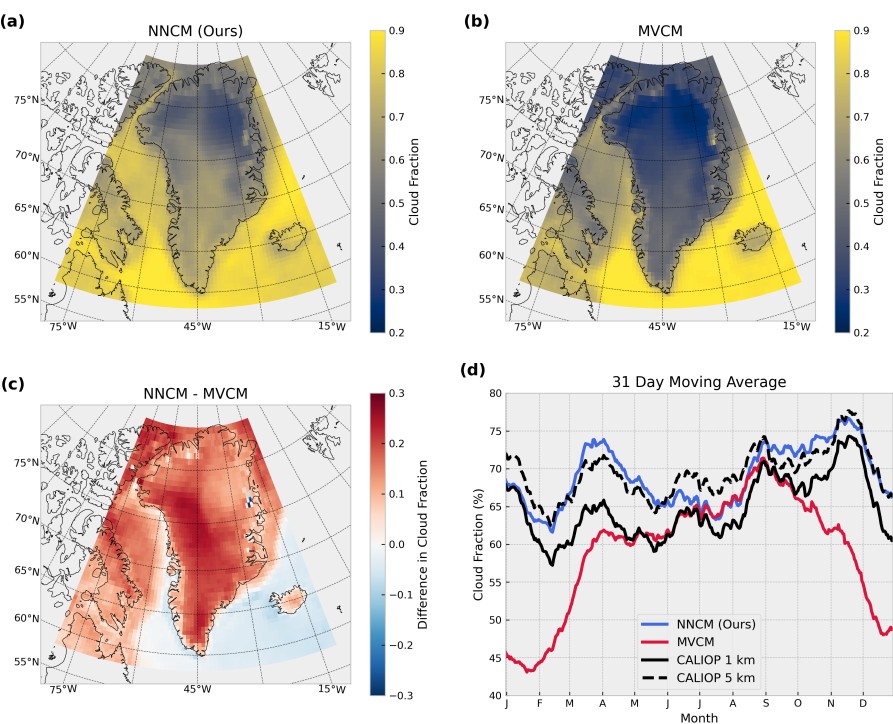

**Figure 13.** Regional analysis of cloud fraction over Greenland. (a) and (b) illustrate the mean cloud fraction for the NNCM and the MVCM for all selected VIIRS scenes in 2019. (c) is the difference between (a) and (b). (d) is the domain-wide 31-day moving average of grid points spatially matched with CALIOP (see main text for details).