# Peer review of "Evaluation of VIIRS Neural Network Cloud Detection against Current Operational Cloud Masks"

_Atmospheric Measurement Techniques, 2020_

## Referee Comment (RC1) · Anonymous Referee #3 · 23 Nov 2020

Review of "Evaluation of VIIRS Neural Network Cloud Detection against Current Operational Cloud Masks"

This Manuscript describes a new methodology to estimate cloud mask using the observations of VIIRS instruments. This method use artificial intelligence tools such neural networks to perform the distinction between pixels contaminated with clouds and the pixels without clouds. Three years of CALIPSO/CALIOP data are used to train, validate and test the neural network. Then, the authors compare the performances of their methods with two other one. Hence, this study shows that neural network tools are adapted to retrieve cloud mask in passive remote sensing with good consistency with active remote sensing. The work is well presented and worthy of publication in Atmospheric Measurement Techniques after answering one major comment and some minor modifications.

Major Comment:

What is exactly, is given in the input of the main neural network? Is it radiances, or reflectances and brightness temperature? For each radiances, is it the differences with climatology (or simulations) and the observation, or raw radiances?

My concern is that the neural network description lack about the physic that is behind such the nature of the input. Also that important information are dispatch in all the study to explain fairly some results, but they are still necessary to be mentioned in the neural network description.

This is more a thought for the conclusion: How does neural network methods will react in the context of global warming and the fast modification of some surfaces? Does it mean that despite the benefits of the accuracy that provide neural network, they are countered by the fact they will need regular updates?

Minor modifications

It would help to provide a table of the VIIRS band.

Page 4, section 2.2: You mentioned in the discussion section (page 16, line 505) that you use ancillary data. But it is poorly described in this section 2.2 (linked to my major comment).

Page 6, lines 171-181: The second part of the section "3.1 Pseudo-Labelling Procedure" is hard to understand at some points. In this section, it is about the neural network that help to account for sun glint. What information is provided

by this neural network to detect sun glint? Is this information provided to the main neural network to not perform a cloud mask, or does it simulate input that are supposed to appear in sun glint condition for the main neural network? Where comes from the information of true sun glint conditions, to be reproduced? Why the 15$^{th}$ day of every month in 2018?

Page 7, line 202: what is the meaning of "binary cross-entropy"?

Page 10, lines 279- 284: Seeing the Figure 4, the difference between TPR of MVCM and the one for neural network is really small. It is most likely that their performances for low broken clouds are similar.

Page 10, line 314-322: In Figure 6, the cloud mask with neural network is less sensitive to variation of latitudes.

Page 11, line 323: "All the of the previous" A word is missing!

Page 12, lines352-354: "This is surprising … a land or water surface." This is really important information it should be mentioned in the description of the neural network input section. (major comment)

Page 14, line 428: "is subject **to** a large"

Page 14, line 447-page 15, line 459: I suggest you put this section and Figure 12, with the section "3.1Pseudo-Labelling Procedure", as it makes the understanding of the pseudo-labelling more clear. Also, because this section is quiet independent of all the analysis of the neural network performances.

Page 23, Figure 1: This paper would benefits of a better scheme that describe the neural network. Better description of the input vector with geo-localisation information. In relation with section 3.2 page 6 and 7, can you say more about the meaning of dropout X% between each layer?

Page 34, Figure 12: There are obvious difference in the behave of the cloud mask from neural network without pseudo-label and the one with pseudo-label. The second cloud mask is more "binary" (i.e. values equal to 0 or 1) than the first one. Can you comment this result? Which neural network of figure 12, have you compared during your paper?

---

## Author Comment (AC1) · 1 Dec 2020

**Response to Reviewer #3 Comments**

*R3: Major Comment: What is exactly, is given in the input of the main neural network? Is it radiances, or reflectances and brightness temperature? For each radiances, is it the differences with climatology (or simulations) and the observation, or raw radiances?*

*My concern is that the neural network description lack about the physic that is behind such the nature of the input. Also that important information are dispatch in all the study to explain fairly some results, but they are still necessary to be mentioned in the neural network description.*

AR: In total, there are 20 different inputs to the cloud masking neural network. The first 16 inputs are from the moderate resolution channels of VIIRS (reflectances for the visible channels and brightness temperatures for the infrared channels). The last four inputs are a binary land/water mask, absolute value of latitude, solar zenith angle, and sun glint zenith angle. We state this in section 2.2, but we think a table would perhaps be more useful here and would facilitate the reader referencing it later on in the paper.

Only the raw observations are used, but we believe some confusion on this point might be coming from statements we make on lines 202-203 where we state that the inputs are standardized by subtracting the mean and dividing by the standard deviation. Scaling inputs is common practice for neural networks, and other approaches are often used such as making the minimum and maximum values 0 and 1, or -1 and 1. If this standardization is not done before training a neural network, then gradients of larger parameters may tend to have larger influence during the training process. In other words, we are rescaling the inputs to the neural network so that each variable has zero mean and unit variance. Otherwise, updates to the neural network may tend to favor the use of inputs with larger values or variance. This is particularly a concern for us since our inputs have very disparate scales: typically between 0 and 1 for reflectances, and between 180 K and 340 K for brightness temperatures.

From one perspective, this might be seen as a framing the inputs as differences from climatology. However, we make no effort to ensure even representation from different seasons or times of day. Similarly, we don't perform this separately for different locations. As a result, we don't believe calling these inputs differences from climatology would be appropriate. Rather, we would simply call them observations rescaled with zero mean and unit variance.

*R3: This is more a thought for the conclusion: How does neural network methods will react in the context of global warming and the fast modification of some surfaces? Does it mean that despite the benefits of the accuracy that provide neural network, they are countered by the fact they will need regular updates?*

This is an interesting, but difficult to answer question! We can think of a few ways in which climate change could affect the performance of this approach. One obvious scenario, which we somewhat touch on in this paper, is the declining presence of arctic sea ice. Evidenced by Figure 10, the cloud masks in this paper can have large TPR differences over different geographical regions, time of day, and surface type. Based on Figure 2, Figure 10, and previous evidence (Liu et al., 2010), we suspect the MVCM underestimates cloud cover over sea ice. As a result, we expect that the decrease in Arctic sea ice, and corresponding increase in ice-free ocean would result in a change

in cloud cover estimated by the MVCM even if it did not result in a real change in cloud cover. This is obviously undesirable, and lends us to believe that cloud detection ability should be invariant as possible to changes in surface type. We believe that Figure 10 illustrates that the neural network may be the least susceptible to this, although the ECM and MVCM are certainly competitive with it in certain regions/conditions.

We suspect that another way in which cloud masks could be impacted by climate change would be a change in the global or regional cloud fraction. The neural network cloud mask is ultimately a statistical model. Under uncertain conditions, it will tend to predict the majority class (usually cloudy) since this is what gives it the best results during training (as measured by binary cross-entropy in our case). We have tried to ensure our approach does not depend too heavily on the use of the background mean cloud fraction by using class-balanced metrics like BACC in our evaluation. Along this same line of thinking we have limited to the amount of geographical information to the absolute value of latitude and a land/water mask. Not including longitude, or more specific surface types was an intentional choice in an effort to reduce on the model's reliance on how the mean cloud fraction varies with this information.

Regular updates would certainly be beneficial, but are not necessarily a specific requirement of our approach. If climate change were to change something fundamental about the decision boundary between clear-sky and cloudy scenes, then all statistical cloud detection models would be impacted. We hypothesize that drifts in sensor calibration or changes in noise levels would be the largest factor in whether a cloud detection model would need regular updates if used in climate data records.

However, it might be interesting to investigate whether machine learning models trained during periods that are dominated by a particular phase of a climate oscillation generalize well to years of the opposite phase. For example, we could imagine issues stemming from approaches trained only during a strong positive ENSO year that utilize longitude and SST as predictors. Assessing how much of an impact this would have would require a very careful experimental setup, and is perhaps specific to what a model uses as predictors and what exactly it is tasked with predicting.

This is certainly something we will be thinking about if we go forward with this approach for future cloud amount analyses. We will plan to add some of this information in our discussion.

*R3: Minor modifications*

*It would help to provide a table of the VIIRS band.*

Agreed. We will plan to add a table like this to our paper.

*R3: Page 4, section 2.2: You mentioned in the discussion section (page 16, line 505) that you use ancillary data. But it is poorly described in this section 2.2 (linked to my major comment).*

Yes. We could certainly do a better job of describing the inputs to the neural network. We think some confusion might be come from our use of the word "ancillary" which we intended to mean an input that is not solely dependent on the VIIRS observations. Using this definition, the landwater mask would be our only ancillary data in the final neural network model since it is a derived product. We will try to clarify this in the text, and add context to our usage of "ancillary," or remove it entirely since its meaning could be ambiguous in the manuscript's current form.

Section 2.2. is another section that we think could benefit from a table. We will plan to include one here that lists which inputs are used for the three neural networks: (1) The main neural network which we are proposing and evaluating the performance of, (2) the pseudolabeling model which only uses infrared channels without solar contributions and provides labels to the main neural network in sun glint scenes, and (3), a neural network that is not trained with pseudolabels and is only used in Figure 12.c to demonstrate the usefulness of pseudolabeling in this context.

*R3: Page 6, lines 171-181: The second part of the section "3.1 Pseudo-Labelling Procedure" is hard to understand at some points. In this section, it is about the neural network that help to account for sun glint. What information is provided by this neural network to detect sun glint? Is this information provided to the main neural network to not perform a cloud mask, or does it simulate input that are supposed to appear in sun glint condition for the main neural network? Where comes from the information of true sun glint conditions, to be reproduced? Why the 15th day of every month in 2018?*

AR: We agree that this text needs to be revised and rewritten, particularly lines 171-181. We think it could benefit from the table in the above response. We could also add a separate table of VIIRS/CrIS fusion channels rather than listing them in the text.

The reason we need the pseudolabeling model, is that we do not have any labels in regions with sun glint where the visible VIIRS channels may give the false impression of cloudiness. To fix this we train a pseudolabeling model that only uses VIIRS IR channels, VIIRS/CrIS fusion channels without solar contributions, latitude, and the land/water mask. The VIIRS/CrIS Fusion channels are used in the pseudolabeling model in an attempt to make up for some of the information lost by removing the visible channels. We removed solar zenith angle and the glint zenith angle since they would not provide useful information for the pseudolabeling model. Aside from these differences, the pseudolabeling model is trained using the same collocations as the main neural network model.

The pseudolabeling model is then used to make predictions in scenes with substantial sun glint. Our determination of substantial sun glint is somewhat subjective, and consists of images with sun glint zenith angles less than 40 degrees. Below 40 degrees, is roughly were we identified visible reflectances starting to increase due to specular reflection and thus, where we would need pseudolabels. The predicted probabilities from the pseudolabeling model are then treated as true labels (as if they were obtained from CALIOP). Essentially, the pseudolabeling model creates cloudy/cloud-free labels that incentivize the main neural network model to not misconstrue high visible reflectance for cloud cover in sun glint scenes.

Scenes from 2018 are selected since that is a year that is included in our training dataset. If 2017 or 2019 were used, our training dataset for the final neural network model would not be independent of the validation and testing datasets. The choice of using the 15th day of each month from 2018 is also somewhat arbitrary. We needed to capture the annual variability of sun angle with respect to latitude, so selecting one day from each month was preferable to selecting 12

consecutive days, for example. We could use more or fewer days, but even after substantial subsampling of these scenes, we found we had more than enough pseudolabels to work with.

*R3: Page 7, line 202: what is the meaning of "binary cross-entropy"?*

AR: Training a neural network requires a loss function (usually called *cost* or *error* function in meteorology and other disciplines) that is differentiable and that one is typically aiming to minimize throughout training. In binary classification problems we usually select binary cross-entropy as our cost function. We will add the equation for the binary cross entropy function to clarify this.

$$J(y, \hat{y}) = -(\, y \log(\hat{y}) + (1 - y) \log(1 - \hat{y}))$$

*Here, J* is the binary cross-entropy loss, $y$ is the binary label from CALIOP, and $\hat{y}$ *is* the predicted cloud probability from the model.

*R3: Page 10, lines 279- 284: Seeing the Figure 4, the difference between TPR of MVCM and the one for neural network is really small. It is most likely that their performances for low broken clouds are similar.*

AR: Agreed. Our phrasing here was partially motivated by the fact that the VIIRS/CALIOP collocations often do not characterize these clouds well since they can often be smaller than the resolution of the 1 km CALIOP Cloud products. Even though the differences in TPR according to the unfiltered VIIRS/CALIOP collocations in Fig 4 are very slight, they may be indicative of a larger overall difference making this result more significant. We do mention this already in the discussion, so we will amend the text here to say the performance here for the collocations is similar, but clarify this expectation later in the discussion.

*R3: Page 10, line 314-322: In Figure 6, the cloud mask with neural network is less sensitive to variation of latitudes.*

AR: Indeed. This is a good point that we will plan to add here.

*R3: Page 11, line 323: "All the of the previous" A word is missing!*

AR: This should read "All of the previous analyses [...]." We will fix this in the final version.

*R3: Page 12, lines352-354: "This is surprising ... a land or water surface." This is really important information it should be mentioned in the description of the neural network input section. (major comment)*

AR: We do mention this in section 2.2, but again, it could be more clear and would certainly benefit from a table describing inputs.

*R3: Page 14, line 428: "is subject to a large"*

AR: Thanks! Yes, this is how that line should read (the word "to" was missing in the original manuscript).

R3: Page 14, line 447-page 15, line 459: I suggest you put this section and Figure 12, with the section "3.1Pseudo-Labelling Procedure", as it makes the understanding of the pseudo-labelling more clear. Also, because this section is quiet independent of all the analysis of the neural network performances.

AR: Agreed. In the current manuscript, there are 8 pages of text separating the description of pseudolabeling and seeing the actual impact of pseudolabeling. This is obviously not ideal. We will make this change in the final version and move Figure 12 and its accompanying text to the end of section 3.1.

R3: Page 23, Figure 1: This paper would benefits of a better scheme that describe the neural network. Better description of the input vector with geo-localisation information.

AR: Absolutely. We will plan to give a more detailed description of the neural network and its input. Figure 1 might also be more effective as a simple table describing the input size, output size, and type of each layer.

R3: In relation with section 3.2 page 6 and 7, can you say more about the meaning of dropout X% between each layer?

AR: Yes. Dropout is a very simple regularization method used in neural networks that helps prevent overfitting. In short, dropout sets a specified number of intermediate activations to a value of zero. Our first layer is a fully-connected layer with 200 units. Dropout(2.5%) indicates that 5 of these 200 units are randomly selected at each training step and set to a value of zero. This helps prevent the model from relying too heavily on any one connection. In our case, we suspect that if we set the dropout rate too high (5% or above) the model had trouble learning relevant features since it was performing worse on our validation dataset during hyperparameter tuning. When we completely removed dropout (by setting it to 0%), the model also performed worse. A dropout rate of 2.5% turned out to be ideal in preventing some amount of overfitting, while still allowing the model to learn relevant features. We will add a sentence or two in the text summarizing this.

R3: Page 34, Figure 12: There are obvious difference in the behave of the cloud mask from neural network without pseudo-label and the one with pseudo-label. The second cloud mask is more "binary" (i.e. values equal to 0 or 1) than the first one. Can you comment this result? Which neural network of figure 12, have you compared during your paper?

AR: Yes. The model that we are analyzing the performance of in the results section is Figure 12.d (the neural network with pseudo-labels). Figure 12.c is shown simply to illustrate how poorly a machine learning model can perform if we don't account for obvious deficiencies in the training dataset (such as a dataset without any sun glint examples).

On the question of the differences between the two models – that is tough to answer for a couple of reasons. First, neural networks are notoriously difficult to interpret the predictions of, and

secondly, sun glint scenes are out-of-domain predictions for the neural network without pseudolabels. Said differently, it is undefined behavior for this model so attempting to interpret its predictions here is even more difficult than usual.

Of course, as we mention in the paper, the neural network without pseudolabels makes erroneously cloudy predictions throughout the entire scene. This is likely because the model without pseudolabels has likely learned to associate high visible reflectivity over water with cloudy pixels since it has never seen sun glint.

All that being said, we can speculate on some of the reasons why Fig 12.d might be more "decisive" than the model in Fig. 12.c. For Fig 12.c there are likely two competing factors: a high visible reflectivity which usually indicates cloudy pixels, and warm infrared brightness temperatures which usually indicates clear pixels. These two factors could result in uncertain conditions since these pieces of information are somewhat contradictory. Fig 12.c is likely decisive because it is making predictions in a sun glint scene where it is exclusively trained with pseudolabels. Rather, than the probabilities being accurate assessments of uncertainty with respect to the CALIOP label, the probabilities in Fig 12.d likely portray the model's ability to accurately reproduce predictions made from a model that exploits solely IR information. To that end, the cloud probability/uncertainty estimates in sun glint regions are not especially useful (which we mention in lines 531-535), but we can verify the actual predicted labels from Fig 12.d appear reasonable compared to the other operation models.

Below we have created drafts of three tables that we will plan to include in the revised version of this manuscript. These tables are aimed at addressing many of reviewers #3 concerns on the nature of the inputs to the neural network. In addition to Table 3, we will include clarifying text that details the purpose of each model, and specifically where each model is used in this work.

| Band | Spectral Range (µm) | Units |
|------|---------------------|-------|
| M1 | 0.400 – 0.421 | Refl. |
| M2 | 0.436 – 0.451 | Refl. |
| M3 | 0.477 – 0.496 | Refl. |
| M4 | 0.541 – 0.561 | Refl. |
| M5 | 0.662 – 0.680 | Refl. |
| M6 | 0.738 – 0.752 | Refl. |
| M7 | 0.843 – 0.881 | Refl. |
| M8 | 1.225 – 1.252 | Refl. |
| M9 | 1.368 – 1.383 | Refl. |
| M10 | 1.571 – 1.631 | Refl. |
| M11 | 2.234 – 2.280 | Refl. |
| M12 | 3.598 – 3.791 | BT [K] |
| M13 | 3.987 – 4.145 | BT [K] |
| M14 | 8.407 – 8.748 | BT [K] |
| M15 | 10.234 – 11.248 | BT [K] |
| M16 | 11.405 – 12.322 | BT [K] |

Table 1: Shown are the names, and spectral ranges of each moderate resolution VIIRS channel. Also shown are the units and whether the channels are expressed in reflectance (Refl.) or Brightness Temperature (BT).

| VIIRS/CrIS Fusion Channel | Spectral Range of MODIS Equivalent Channel (µm) |
|---|---|
| MODIS 27 | 6.535 – 6.895 |
| MODIS 28 | 7.175 – 7.475 |
| MODIS 29 | 8.400 – 8.700 |
| MODIS 30 | 9.580 – 9.880 |
| MODIS 31 | 10.780 – 11.280 |
| MODIS 32 | 11.770 – 12.270 |
| MODIS 33 | 13.185 – 13.485 |
| MODIS 34 | 13.485 – 13.785 |
| MODIS 35 | 13.785 – 14.085 |
| MODIS 36 | 14.085 – 14.385 |

Table 2: Shown are the names of each infrared VIIRS/CrIS Fusion channel that is used in the pseudo-labeling model. The stated spectral ranges are those of the MODIS equivalent channels. All channels are expressed as brightness temperatures [K].

| Inputs | Neural Network with Pseudo-labels | Neural Network without Pseudo-labels | Pseudo-labeling Model |
|---|---|---|---|
| M1-M11 | X | X | |
| M12-M13 | X | X | |
| M14-M16 | X | X | X |
| MODIS27-MODIS36 | | | X |
| \| Latitude \| | X | X | X |
| Solar Zenith Angle | X | X | |
| Sun Glint Zenith Angle | X | | |
| Land/Water Mask | X | X | X |

Table 3: Summary of the information used by each neural network model in this work.

Once again, we would like to express our thanks to the reviewer for volunteering their time to give us feedback on our manuscript. They have very helpfully identified several areas in which we can improve the quality, presentation, and clarity of this work.

---

## Referee Comment (RC2) · Anonymous Referee #2 · 3 Dec 2020

**1  General comments**

The paper compares a neural network cloud mask trained with 2D features to two operational cloud masks. The algorithm is trained with CALIOP data and uses a pseudo labeling method to deal with the issue that sunglint areas are not covered by the co-location dataset. The neural network cloud mask uses a large network but few sources of traditionally used ancillary data (most notably surface temperature is not included). Performance is very good except for small broken clouds; which given the 2D features is a bit counter-intuitive. The same network gives similar results for a large variety of surfaces.

[Figure]

**2 Specific comments**

1. The introduction is missing an important reference. The ESA cloud CCI algorithm also uses a neural network trained with CALIOP data for the cloud mask but with a different network structure, training, imager etc.

2. Line 61-65: *Our approach aims to improve upon existing literature in several ways. Rather than relying on precomputed spectral, or textural features, we allow a neural network to learn relevant features from a local 3 pixel by 3 pixel image patch from all 16 moderate resolution VIIRS channels.* The section is missing a motivation as to why it might be good to let the network learn the relevant feature itself. If the relevant features can be precomputed then the network can be made smaller and faster (fewer variables, fewer layers).

3. Line 70-75: Include short descriptions of the MVCM and ECM cloud mask methods. (Does not have to be here.)

4. Line 185-210: Did you use any available software for training the network?

5. Line 234: Could the slightly overestimated cloud fraction in day time for MVCM be due to thin clouds not detected by 1km CALIOP data, but detected in the 5km CALIOP data and the MCVM? The chance of detecting these very thin clouds should be larger during day time.

6. Table 1: Add also at least TPR, TPN and CALIOP cloud fraction to a table. It is the 2019 data that is used for the table, right? Add info in the caption. Include also a table with results for the unfiltered data.

7. Line 265: I find this surprising, I would have expected the 2D feature to be most useful for fractional clouds.

8. Line 306: You mention that Bayesian algorithms might be affected by climatological means. Considering that your method includes latitude could it not be that it too uses the latitude mean cloudiness from the two years of training data? Have you tested how much the network depends on latitude?

9. Line 357: Can the latitude combined with sun zenith angle give a rough estimate of surface temperatures? Very impressive results for temperatures close to surface temperature.

10. Line 390: I agree it is not bad with a consistent TPR dependent only on the cloud. But optimizing TPR differences might mean making the TPR lower in easy conditions to match the performance in more difficult conditions. Is it not equally important to keep TPN as constant as possible? I think this is what is more traditionally aimed at.

11. Line 425: For the validation data do you have sea ice cover to the north west of Greenland? Can the shrinking sea ice cover in the arctic be part of the explanation. If MVCM is trained on older data and assumes it to be sea ice, and the new NN approach trained on more recent data expects more water?

12. Line 433: *The averages across space are weighted by the cosine of latitude expressed in radians.* I do not understand what you mean here.

13. Line 457: *The pseudo-labeling model likely has low skill in such conditions due to the low contrast between a low-level fractionally cloudy pixel and the background.* Did you consider using the ECM for the pseudo-labeling?

14. Line 537: *Additionally, we have not evaluated how the neural network performs specifically in cloud-free scenes with high aerosol loading. We expect that this could depend largely on the ability for CALIOP to distinguish cloud from aerosol layers.* Even if it does depend on CALIOP's ability should it not depend mostly on the VIIRS capabilities?

[Figure]

15. Figure 6: Consider adding figures also for the BACC.

16. Figure 8: Is this filtered or unfiltered data?

17. Have you tested to applid the NN on older data (2013) and was there a difference in performance?

18. Is execution time comparable with the operational cloud masks? Is it feasible to use for nowcasting?

19. A name of the method would be useful.

20. From my experience with NN cloud masks results often look less realistic close to the swath edges when comparing results to the RGB. In Figure 12 results look realistic also closer to the edges. Is this normally the behavior?

**3  Technical corrections**

- Line 53: Häkansson should be Håkansson (several places)

---

## Referee Comment (RC3) · Anonymous Referee #4 · 7 Dec 2020

A new neural network cloud mask for VIIRS measurement is presented. The neural network is trained with collocated CALIOP observations. Using a global testing dataset of one year, the performance of the neural network is evaluated, using several metrics, for different categories like land/water, day/night, latitude range and varying COT threshold. Results show general good agreement with mean cloud fraction from CALIOP, including consistency between different categories, though larger differences are found for small-scale, low-level clouds. Comparison to two operational VIIRS cloud mask show that the neural network outperforms them for almost all conditions, however also here the struggle with small-scale, low-level clouds is evident. The largest improvements are found for collocations at higher latitudes.

[Figure]

In general the manuscript is well structured. The method is clearly presented, including many corresponding references , and considerations made during the set-up of the neural network well explained. The manuscript could benefit from some additional information on the data used as well as from more details on the two operational cloud masks. The assessment of the performance and comparisons are done in multiple ways and accompanying figures clearly presented and explained. Issues and differences are analyzed and extensive discussion provided.

Minor comments/questions:

Line 85: Before going straight to the Collocation Methodology I would recommend to add a small subsection on the VIIRS instrument/observations as well as for the two operational cloud mask swith which a lot of comparisons are done.

Line 86: Also some more information on the CALIOP data could be provided, like what is the width of one cloud layer, time of overpass etc.?

Line 105: Would be nice to see a global map of sampling frequency of valid collocations for the training dataset, maybe even per season, like is presented for the testing dataset (Fig. 6 b).

Line 108: Observations in form of radiances/brightness temperatures? Please provide more detail on the input for the neural network.

Line 109: How are the eight categories combined?

Line 175: It is not clear to me how the sun glint scenes are labeled, on a pixel-basis? There is a reference, but some more information would be nice.

Line 202: All inputs are standardized.. meaning for the 3 x 3 pixels?

Line 217: Already refer to corresponding equation numbers.

Line 314: Why not continue with BACC?

Line 351: How are the surface temperature from the model matched, spatially and temporally, with the measurements? Some more detail should be provided.

Line 354: are smaller than

Line 360/Fig 6.: The large negative difference for the grid cell in front of the coast of Namibia, could that be related to biomass burning aerosol layers?

Line 472: Could some (pseudo) labeling technique be useful here? Or using a larger pixel matrix than 3 x 3? Maybe combined with taking information from not only 1 CALIOP profile but from adjacent profiles as well?

Technical corrections Line 29: ..large amounts of training data.. Line 47: ..how a very simple.. Line 298: .. compared to the MVCM.. Line 323: All of the previous.. Line 325: ..depend on the particular.. Line 363: the distribution of the.. Line 423: .. may be a result of sea ice cover. Line 428:.. is subject to a large amount..

---

## Author Comment (AC2) · 22 Dec 2020

*R2: 1 General comments The paper compares a neural network cloud mask trained with 2D features to two operational cloud masks. The algorithm is trained with CALIOP data and uses a pseudo labeling method to deal with the issue that sunglint areas are not covered by the colocation dataset. The neural network cloud mask uses a large network but few sources of traditionally used ancillary data (most notably surface temperature is not included). Performance is very good except for small broken clouds; which given the 2D features is a bit counter-intuitive. The same network gives similar results for a large variety of surfaces.*

*2 Specific comments*

*1. The introduction is missing an important reference. The ESA cloud CCI algorithm also uses a neural network trained with CALIOP data for the cloud mask but with a different network structure, training, imager etc.*

AR: Thank you for pointing this out. This work is highly relevant to our paper. We will plan to include reference to this in the final version.  Below seems to be the appropriate citation for the work mentioned (but please correct us if not):

Sus, O., Jerg, M., Poulsen, C., Thomas, G., Stapelberg, S., McGarragh, G., Povey, A., Schlundt, C., Stengel, M., and Hollmann, R.: The Community Cloud retrieval for CLimate (CC4CL). Part I: A framework applied to multiple satellite imaging sensors, submitted to Atmospheric Measurement Techniques Discussions, pp. , 2017.

*R2: 2. Line 61-65: Our approach aims to improve upon existing literature in several ways. Rather than relying on precomputed spectral, or textural features, we allow a neural network to learn relevant features from a local 3 pixel by 3 pixel image patch from all 16 moderate resolution VIIRS channels. The section is missing a motivation as to why it might be good to let the network learn the relevant feature itself. If the relevant features can be precomputed then the network can be made smaller and faster (fewer variables, fewer layers).*

AR: Yes, we have not properly motivated this statement. It is also our experience that supplying precomputed to the neural network results in a network with fewer parameters after hyperparameter tuning. Both the ECM and the MVCM make use of cloud tests that involve some amount of feature engineering. In this paper we are starting with the assumption that we don't know all the relevant features for cloud detection. Despite the considerable amount of work has been done in the development of both the operational cloud masks, there might still be variability among the imager channels that is relevant to cloud detection and currently going unexploited (particularly that which involves three or more channels).

*R2: 3. Line 70-75: Include short descriptions of the MVCM and ECM cloud mask methods. (Does not have to be here.)*

AR: We agree that short descriptions of each should be added and will plan to do so. This will hopefully help readers understand differences between the methods that we write about later in the discussion section.

*R2: 4. Line 185-210: Did you use any available software for training the network?*

AR: Yes, we used the numpy, tensorflow, and keras python libraries. We will plan to add reference to them in the final version.

*R2: 5. Line 234: Could the slightly overestimated cloud fraction in day time for MVCM be due to thin clouds not detected by 1km CALIOP data, but detected in the 5km CALIOP data and the MCVM? The chance of detecting these very thin clouds should be larger during day time.*

AR: This is a great point. The 1km CALIOP Cloud products are less sensitive to optically thin cloud cover than the 5 km product, and it's not unreasonable to expect that one of the operational masks might detect clouds missed by the 1km product, but correctly identified by the 5 km product. We will plan to describe this in the results or discussion section.

*R2: 6. Table 1: Add also at least TPR, TPN and CALIOP cloud fraction to a table. It is the 2019 data that is used for the table, right? Add info in the caption. Include also a table with results for the unfiltered data.*

AR: We are assuming that by TPN, the reviewer means the TNR (the True Negative Rate). If this assumption is wrong, please correct us so we can properly address your comment.

We will remake this table with BACC, TPR, TNR and cloud fraction. It is indeed for the 2019 testing dataset. We did not initially include a table for the unfiltered dataset because it did not offer much additional insight for earlier versions of our model that wasn't apparent in our other tables and figures. However, if we don't include such a table, it could be interpreted as an obvious oversight since earlier in the text we emphasize the importance of the differences between the unfiltered and filtered datasets. At the very least we will plan to include the unfiltered table in the supplement. We will consider the unfiltered results more carefully before the final version and will move it to the main text if there are interesting differences to discuss.

*R2: 7. Line 265: I find this surprising, I would have expected the 2D feature to be most useful for fractional clouds.*

AR: We agree that this result is a bit counter-intuitive. We believe this is a limitation of using CALIOP as our source of labeled data. It is likely that small broken clouds are not well represented in our collocation database due to the size of these clouds and the time difference between when the two instruments observe them. Small horizontal displacements of these clouds between times that both VIIRS and CALIOP observe the same ground location could be mean that some of our labels for these types of clouds are more prone to error, and result in poor characterization of them from our neural network approach.

*R2: 8. Line 306: You mention that Bayesian algorithms might be affected by climatological means. Considering that your method includes latitude could it not be that it too uses the latitude mean cloudiness from the two years of training data? Have you tested how much the network depends on latitude?*

AR: Thanks for this comment! Very early on in this work we were using models that were somewhat more interpretable than the neural network we talk about in this paper (Gradient Boosting Machines). We found that including latitude changed how other features were used in very interesting ways. For example a difference between the 11µm and 8.6µm brightness temperature of 1 K had almost no impact at the equator, but was one of the most influential features in the model at the high latitudes. Of course, the neural network is a completely different model and there is absolutely no guarantee that usage of latitude is similar here, but this was our initial motivation for including it.

Motivated by this comment, we retrained our neural network without latitude. Surprisingly, the daytime results improved in our testing dataset by very small margins. The surface type with the largest difference during the daytime was (non-permanent) snow which increased from 95.5% to 95.8% using the filtered dataset. During the night, removing latitude overall worsened our results slightly more substantially. The largest change was, again, over (non-permanent) snow which decreased from 92.0% to 90.9%.

We additionally retrained the model a second time after removing latitude, solar-zenith angle, and sun glint zenith angle. This is because the distribution of solar zenith angle and sun glint zenith angle vary with latitude, and could leak some information to the model about the latitudinal mean cloudiness (low sun glint angles typically occur at low latitudes, for example).

This model had very slightly worse results in the daytime than the model without latitude (the largest difference was -0.2%). The global nighttime BACC changed from 93.4 to 92.9 with most of the change coming from nighttime water surfaces with changed 93.2 to 92.4. The other surface types remained unchanged with differences within +/- 0.1%.

Overall, the model seems to mostly worsen in nighttime water scenes when removing latitude and information related to latitude. Considering these results, I think it is likely that our model depends on latitudinal mean cloudiness in some capacity over these areas. However, it is difficult to quantify whether it is serving a purpose similar to that of a climatological mean, or if it is changing the usage of other observations features (like we have previously observed in our other models).

*R2: 9. Line 357: Can the latitude combined with sun zenith angle give a rough estimate of surface temperatures? Very impressive results for temperatures close to surface temperature.*

AR: This is a really great point that we did not think about. Even though we have not provided surface temperature directly to the model, perhaps the other information we have provided is utilized in a similar way that a mean surface temperature would.

I built a model that uses solar zenith angle, the land/water mask, sun glint angle, and the absolute value of latitude to predict the GFS surface temperature. It is trained in a similar way as the model in the manuscript but with fewer parameters. It has 3 fully connected layers of 20, 10 and 1 units, no dropout, a linear activation as the last layer, and mean squared error as the loss function.

[Figure]

On the 2019 testing dataset, the model produced a mean absolute error of 6.7 K and a mean squared error of 86.3 K. I think it is somewhat possible that this information is serving similar purpose as an extremely rough estimate of surface temperature. However, I do not know if a mean absolute error of 6.7 K is accurate enough to be especially useful in cloud detection.

To add another perspective to this question, I retrained the cloud detection model only using the 16 VIIRS channels and have recreated the thermal contrast figure (Figure 8). The differences in this figure and the one in the manuscript are fairly minimal (except that we added both datasets to address a later comment). The largest difference in performance with and without this information is a change of roughly -3% ACC around -10 to -16 K. Based on these small differences I think it is safe to rule out the idea that this information serves a similar purpose as surface temperature might.

[Figure]

*R2: 10. Line 390: I agree it is not bad with a consistent TPR dependent only on the cloud. But optimizing TPR differences might mean making the TPR lower in easy conditions to match the*

*performance in more difficult conditions. Is it not equally important to keep TPN as constant as possible? I think this is what is more traditionally aimed at.*

AR: (Again we are assuming that TNR was meant instead of TPN, but please correct us if not). I think the usefulness of this metric perhaps depends on the application. There is indeed a tradeoff between performance with respect to CALIOP and cloud detection consistency. For operational nowcasting, we expect that users might not be especially concerned with detection consistency over different surface types or times of day. For climate applications, this might be a more important consideration. For example, a globally uniform increase in the amount of clouds with optical depth of about 0.1 might only be detected as a much larger increase over certain surface types with higher TPR for these specific clouds. This might result depictions of cloud cover change that don't align with reality. Minimizing TNR differences between different conditions could achieve a similar result, we think. TPR differences might be more useful since one could weigh the differences with respect to optical depth. For example, a large TPR difference at a high cloud optical depth is likely more problematic than one at a very low optical depth.

Either way, we recognize that our suggestion that TPR differences should be minimized in conjunction with other performance metrics might be an overstatement. We will think more carefully about this in the final version, and perhaps only suggest that it be used as a metric to identify detection consistency when that is a specific need of a cloud mask.

*R2: 11. Line 425: For the validation data do you have sea ice cover to the north west of Greenland? Can the shrinking sea ice cover in the arctic be part of the explanation. If MVCM is trained on older data and assumes it to be sea ice, and the new NN approach trained on more recent data expects more water?*

AR: Great question. Our understanding is that the MVCM uses separate decision pathways based only surface type and time of day. Quoting from the MVCM users guide (Frey et al. 2019): "Several ancillary data sets serve as inputs to the MVCM process, [...] Near-Real-Time SSM/I-SSMIS (Special Sensor Microwave/Imager-Special Sensor Microwave Imager/Sounder) EASE (Equal-Area Scalable Earth)-Grid Daily Global Ice Concentration and Snow Extent (NISE) files contain daily global gridded snow and ice extent." Assuming the correct up-to-date surface type is given to MVCM, we do not expect this to be an issue.

However, this does bring up another an interesting point: What if the surface type or surface temperature is not perfectly known? If the operational methods assume incorrect characteristics of the surface, then this could be a reason why our approach appears to perform better over scenes where the $11\mu m$ BT is close to the NWP surface temperature. Similarly, if the presence of sea-ice is incorrectly indicated, this might be problematic for approaches that otherwise depend on this information.

*R2: 12. Line 433: The averages across space are weighted by the cosine of latitude expressed in radians. I do not understand what you mean here.*

AR: Yes, this sentence is poorly-worded. When doing this analysis we compute the values on a grid with regular latitude/longitude spacing. Because of this, high-latitude grid cells represent a

smaller surface area than the lower-latitude grid cells. To account for this we give a smaller weight to the higher latitude observations when averaging across the domain. To calculate the weights, we first convert the latitude of the grid cells to radians, and then take the cosine of those values. We will revise this sentence to hopefully make this more clear. Perhaps something like the following: "When calculating the mean cloud fraction, individual values on the regular latitude/longitude grid are weighted to account for differences in surface area between them."

*R2: 13. Line 457: The pseudo-labeling model likely has low skill in such conditions due to the low contrast between a low-level fractionally cloudy pixel and the background. Did you consider using the ECM for the pseudo-labeling?*

AR: We considered it, but did not actually do this analysis. When doing the analysis for this manuscript our goal was to demonstrate the effectiveness of the neural network as a stand-alone approach for cloud detection. It is clear that the ECM is better in very specific scenarios and has obvious advantages of being more interpretable. In practice, it would surely be better to use the ECM for pseudolabeling of sun glint scenes. However, if a cloud mask is needed for a sensor in which the ECM has not been validated for, we wanted to demonstrate how sun glint issues could be mitigated without it

*R2: 14. Line 537: Additionally, we have not evaluated how the neural network performs specifically in cloud-free scenes with high aerosol loading. We expect that this could depend largely on the ability for CALIOP to distinguish cloud from aerosol layers. Even if it does depend on CALIOP's ability should it not depend mostly on the VIIRS capabilities?*

AR: Agreed. We wanted to mention the point that CALIOP's ability to distinguish cloud from aerosol would be an additional complicating factor. We now realize that these sentences, as written, are somewhat misleading. We will plan to rewrite this statement to something like this: "In addition to the challenge of distinguishing cloudy from cloud-free scenes with high aerosol loading using VIIRS measurements, we expect that CALIOP's ability to discriminate between them and provide accurate labels to the neural network model could be another complicating factor."

*R2: 15. Figure 6: Consider adding figures also for the BACC.*

AR: Other reviewers have asked for this as well so it is clear that we should add this figure (at the very least in the supplement if not in the main text). We initially did not include a figure with BACC since there were grid cells where the mean CALIOP cloud fraction was particularly high, and the BACC was mostly dependent on a relatively small amount of cloud-free CALIOP collocations (the Southern Ocean, for example.) Below is the same figure with BACC, and panel (b) is replaced with the CALIOP mean cloud fraction. Similar to the ACC figure in the manuscript, these maps are calculated using the filtered dataset.

[Figure]

*R2: 16. Figure 8: Is this filtered or unfiltered data?*

AR: It was filtered data, but we see no reason by we couldn't include both datasets here. See below for figure. Actually, there are interesting differences in the unfiltered data that that don't exist in the other. For the neural network and the MVCM there is either a global or local minimum around -6 to -8 K. We will plan to revise this figure with both datasets and add some discussion of this.

[Figure]

*R2: 17. Have you tested to applid the NN on older data (2013) and was there a difference in performance?*

AR: Not yet, but we plan to examine something similar to this in future work. Whether or not ML-based approaches generalize to data collected long after (or before, in this case) their training period is an interesting question. This perhaps partially depends on whether our approach learned variability that is specific to our training period (2016 and 2018) which seems clearly undesirable. Of course, consistent sensor calibration is also a significant concern with this.

*R2: 18. Is execution time comparable with the operational cloud masks? Is it feasible to use for nowcasting?*

AR: I wasn't particularly comfortable commenting on execution time in the manuscript since it might vary greatly across implementations or systems. Using a fairly recent GPU processing an entire 6-minute VIIRS scene with our neural network implementation took roughly 7 seconds and only hit roughly 50% utilization of the GPU. To contrast, using 2 threads of a very old (8-10 years?) CPU it took 90 seconds for the full scene and roughly 25 seconds using 12 threads. Nowcasting is certainly a possibility if the machine has access to a GPU or a modern CPU. If running on older hardware, the processing time might be unfortunately prohibitive. That being said, we are definitely not expert software engineers, so others may have more luck in writing fast processing code.

I am not familiar with the MVCM processing code. Based only on the method itself, I suspect the MVCM is the fastest, and that the neural network is by far the slowest if run on similar hardware.

*R2: 19. A name of the method would be useful.*

AR: Agreed. This might help make the text more concise and readable. We will spend some time thinking of an appropriate name, or at least an abbreviation to use in the text.

*R2: 20. From my experience with NN cloud masks results often look less realistic close to the swath edges when comparing results to the RGB. In Figure 12 results look realistic also closer to the edges. Is this normally the behavior?*

AR: Figure 12, specifically, is a cropped image close to the nadir track of VIIRS to focus on the area of sun glint.

Early experiments we performed with MODIS/CALIOP data showed issues with viewing angle, but this is expected due to MODIS only making near-nadir collocations with CALIOP. Håkansson et al. 2018 discuss this as well with cloud-top pressure/altitude.

VIIRS makes collocations with CALIOP at a larger variety of viewing angles (0 to 50 degrees in our dataset). Early on in the development of our model we included sensor zenith angle as a predictor. However, we noticed that performance with respect to CALIOP increased when it was removed so we left it out. Aside from the regional analysis over Greenland, we have not extensively analyzed full scenes to the swath edges aside from a set of 20-30 scenes that we run as a "sanity check". In these scenes we have not noticed particularly unrealistic behavior at swath edges.

*R2: 3 Technical corrections • Line 53: Häkansson should be Håkansson (several places)*

AR: An embarrassing mistake on our part! We will make sure this is fixed in the final version.

---

## Author Comment (AC3) · 22 Dec 2020

R4: *A new neural network cloud mask for VIIRS measurement is presented. The neural network is trained with collocated CALIOP observations. Using a global testing dataset of one year, the performance of the neural network is evaluated, using several metrics, for different categories like land/water, day/night, latitude range and varying COT threshold. Results show general good agreement with mean cloud fraction from CALIOP, including consistency between different categories, though larger differences are found for small-scale, low-level clouds. Comparison to two operational VIIRS cloud mask show that the neural network outperforms them for almost all conditions, however also here the struggle with small-scale, low-level clouds is evident. The largest improvements are found for collocations at higher latitudes.*

*In general the manuscript is well structured. The method is clearly presented, including many corresponding references , and considerations made during the set-up of the neural network well explained. The manuscript could benefit from some additional information on the data used as well as from more details on the two operational cloud masks. The assessment of the performance and comparisons are done in multiple ways and accompanying figures clearly presented and explained. Issues and differences are analyzed and extensive discussion provided.*

*Minor comments/questions: Line 85: Before going straight to the Collocation Methodology I would recommend to add a small subsection on the VIIRS instrument/observations as well as for the two operational cloud mask swith which a lot of comparisons are done.*

AR: Agreed. We will plan to add more details about the VIIRS instrument, and explain the general approaches of the two operational cloud masks.

R4: *Line 86: Also some more information on the CALIOP data could be provided, like what is the width of one cloud layer, time of overpass etc.?*

AR: Agreed. We will plan to add more information about the CALIOP data.

R4: *Line 105: Would be nice to see a global map of sampling frequency of valid collocations for the training dataset, maybe even per season, like is presented for the testing dataset (Fig. 6 b).*

AR: Yes, the distribution of collocations based on time of year and lat/lon is very similar between the testing dataset, but it certainly wouldn't hurt to show it explicitly in a figure.

R4: *Line 108: Observations in form of radiances/brightness temperatures? Please provide more detail on the input for the neural network.*

AR: Agreed. Other reviewers also raised this concern as well so it is clear we need to do a better job of explaining the inputs to the neural network. We will make sure this is improved in the final revised version.

R4: *Line 109: How are the eight categories combined?*

AR: The categories are 0=shallow ocean, 1= land, 2=coastline, 3=shallow inland water, 4= ephemeral water, 5 = deep inland water, 6 = continental/moderate ocean,  7=deep ocean.

We combine these categories to more generally describe land/water classification. In our binary land/water mask (land=1, water=0) the land category is made up of "land" and "coastline" categories of the original mask. All other water/ocean categories of the original mask are combined into a simply a water category. We will give more details about this in the final version

R4: *Line 175: It is not clear to me how the sun glint scenes are labeled, on a pixel-basis? There is a reference, but some more information would be nice.*

AR: Yes. The pseudolabeling model, which is invariant to sun glint, is used to make predictions where sun glint is present. The only differences between our main cloud detection model and the pseudolabeling model are that we include the infrared VIIRS/CrIS fusion channels in the pseudolabeling model, and remove all channels with solar contributions. It similarly accepts a a 3x3 pixel patch from each channel as input, and makes a cloudy/cloud-free prediction for the center pixel. After running this model for every pixel in several scenes with sun glint, we subsample these predictions (since we had more than enough data to work with).

Other reviewers have also suggested revising this section, so it is clear that we need to make some changes to the text in order to more clearly present this information. We will plan to revise this section in the final version of this paper.

R4: *Line 202: All inputs are standardized.. meaning for the 3 x 3 pixels?*

AR: The datasets are standardized using means and standard deviations computed for the entire training dataset, not just for each 3x3 patch. So for M15 (the ~10.8 µm brightness temperature), the one value for the mean is computed from all M15 observations in our training dataset (and the same for the standard deviation). The reason we do this is merely a technical consideration and is common practice for neural networks. We want the inputs to have approximately similar ranges. Inputs with very disparate scales can sometimes cause issues during training for neural networks.

R4: *Line 217: Already refer to corresponding equation numbers.*

AR: Yes. This is the first reference of those abbreviations/metrics so we will fix this accordingly.

R4: *Line 314: Why not continue with BACC?*

AR: Reviewer 2 asked a similar question so we are quoting our response here "We initially did not include a figure with BACC since there were grid cells where the mean CALIOP cloud fraction was particularly high, and the BACC was mostly dependent on a relatively small amount of cloud-free CALIOP collocations (the Southern Ocean, for example.) Below is the same figure with BACC, and panel (b) is replaced with the CALIOP mean cloud fraction. Similar to the ACC figure in the manuscript, these maps are calculated using the filtered dataset." We will plan to add this in the either the supplement or the main text.

[Figure]

R4: *Line 351: How are the surface temperature from the model matched, spatially and temporally, with the measurements? Some more detail should be provided.*

AR: First, we obtain the surface temperature from the 6-hourly 0.5 degree GFS 12-hour forecast (we incorrectly said this was the analysis in the main text – we will fix this) for the file occurring before and the file occurring after the VIIRS scene. Then, linear interpolation is used in time and space to approximate the resolution and time of the VIIRS scene.

One consideration that we did not discuss was the disparity in spatial resolution and differences in time between the VIIRS observations and the GFS 12-hour forecasts. Some of the differences in Figure 8 are perhaps due to discrepancies between the actual surface temperature and what was estimated by GFS due to these issues. Maybe the relatively poor performance the operational models in Figure 8 of the models is not only due to low thermal contrast, but mischaracterization of the surface temperature? We will think about these differences a bit more and add them to the main text in the final version.

R4: *Line 354: are smaller than*

AR: I think we should reword this statement altogether. Originally we were meaning the signed value was greater (in the sense that 4 > -6) but we now realize this is confusing. Maybe we should instead say something along the lines of "The performance of all models decreases as the

VIIRS 10.8µm brightness temperatures become more similar to or larger than the surface temperature.

R4: *Line 360/Fig 6.: The large negative difference for the grid cell in front of the coast of Namibia, could that be related to biomass burning aerosol layers?*

AR: This was what we originally suspected but didn't look into it much further.

We tracked down the poor performance in this grid cell to an individual nighttime scene over the where the neural network achieved an accuracy only a 25% over a stretch of roughly 550 CALIOP collocations over the ocean near the coastline that were 97% cloud-free. To compare, the ECM had 81.5% accuracy and the MVCM has 97.8% accuracy over this same stretch.

The very poor performance of this specific scene appears to be from a number of factors. One was a processing error where the land/water mask was not reduced from the 8 categories to the binary mask (we suspect a job on our cluster was preempted or terminated early).

We checked the closest daytime overpass and adjacent regions had aerosol optical depth of roughly 0.15 to 0.3 estimated by a VIIRS data. Moderate aerosol loading was somewhat apparent in the true color images.

We think the most influential factor was what appeared to be particularly cold SSTs along the coastline indicative of upwelling. Without the visible channels in this nighttime scene, the neural network misconstrued these cold surface temperatures as cloud cover. We suspect this to be a scenario where a rough estimate of surface temperature could improve the accuracy of our approach. After fixing the error in the land/water mask the accuracy over this small stretch of CALIOP collocations improves from 25% to 63% -- still trailing behind the ECM and MVCM significantly.

R4: *Line 472: Could some (pseudo) labeling technique be useful here? Or using a larger pixel matrix than 3 x 3? Maybe combined with taking information from not only 1 CALIOP profile but from adjacent profiles as well?*

AR: Sure! As another reviewer suggested we might be able to pseudolabel using the MVCM or ECM to address specific deficiencies in the neural network. For the purposes of this paper we wanted to demonstrate how the neural network could be implemented in a stand-alone capacity (assuming that neither of the operational masks were available).

We briefly experimented with a convolutional neural network that used 5x5 and 7x7 patches. These models actually performed slightly worse for broken clouds and only improved performance in more homogenous scenes (represented by the filtered dataset).

Using adjacent CALIOP profiles could be interesting, but we would have to think carefully about how this might work. We noticed in previous experiments when we trained to the 5 km CALIOP product that resulting cloud mask was extremely "smooth" and did not at all capture the finescale variability of scenes with broken clouds. We worry that using information from adjacent profiles might lead to a similar effect.

This might be a situation where manual labeling could be a good option. One way to do this efficiently could be to find regions of broken clouds in VIIRS images, draw a bounding box around such regions, and choose a threshold on the 11µm channel that reliably separates cloud-free from cloudy pixels (using a threshold specific to each manually selected region). This is, of course, very subjective.

R4: *Technical corrections Line 29: ..large amounts of training data.. Line 47: ..how a very simple.. Line 298: .. compared to the MVCM.. Line 323: All of the previous.. Line 325: ..depend on the particular.. Line 363: the distribution of the.. Line 423: .. may be a result of sea ice cover. Line 428:.. is subject to a large amount..*

AR: Thanks for pointing out these errors! We will make sure they are fixed in the final version.

---

## Author Response (AR1)

**Author Response to Referee Comments**

In the three author comments, we have responded (sometimes at length) to all points in the referees' feedback individually. Below, we have collated all referee comments. For each instance we either quote from our original author comment or otherwise summarize how the comment was addressed in our revised manuscript.

After our responses to each referee comment, we include a log of the changes we have made to the manuscript. We have put a significant amount of effort to ensure this list is accurate, but many small changes (regarding phrasing, typos, and grammer) may have been missed.

At the end of this document we have included the tracked changes that includes the diffs between the original and revised manuscript (created using the latexdiff package). Please note that since tables and figures have been added and reordered, the diffs are not accurate for tables and figures. In that case, please compare the figures included here with those of the original submission. Section 2 has also been renamed to "Instruments and Data", which seems to have thrown off the numbering in the latexdiff document.

**Referee #2 Comments**

**R2:** *The introduction is missing an important reference. The ESA cloud CCI algorithm also uses a neural network trained with CALIOP data for the cloud mask but with a different network structure, training, imager etc.*

**AR:** We have added the following reference: Sus, O., Jerg, M., Poulsen, C., Thomas, G., Stapelberg, S., McGarragh, G., Povey, A., Schlundt, C., Stengel, M., and Hollmann, R.: The Community Cloud retrieval for CLimate (CC4CL). Part I: A framework applied to multiple satellite imaging sensors, submitted to Atmospheric Measurement Techniques Discussions, pp. , 2017

**R2:** *2. Line 61-65: Our approach aims to improve upon existing literature in several ways. Rather than relying on precomputed spectral, or textural features, we allow a neural network to learn relevant features from a local 3 pixel by 3 pixel image patch from all 16 moderate resolution VIIRS channels. The section is missing a motivation as to why it might be good to let the network learn the relevant feature itself. If the relevant features can be precomputed then the network can be made smaller and faster (fewer variables, fewer layers).*

**AR:** We have added details to motivate this choice of our approach.

**R2**: *3. Line 70-75: Include short descriptions of the MVCM and ECM cloud mask methods. (Does not have to be here.)*

**AR:** Descriptions of each mask have been added as well as details about the VIIRS sensor.

**R2:** *4. Line 185-210: Did you use any available software for training the network?*

**AR:** We have added references to the software used.

**R2:** *5. Line 234: Could the slightly overestimated cloud fraction in day time for MVCM be due to thin clouds not detected by 1km CALIOP data, but detected in the 5km CALIOP data and the MCVM? The chance of detecting these very thin clouds should be larger during day time.*

**AR:** Quoting from our author comment: "The 1km CALIOP Cloud products are less sensitive to optically thin cloud cover than the 5 km product, and it's not unreasonable to expect that one of the operational masks might detect clouds missed by the 1km product, but correctly identified by the 5 km product." A brief comment on this has been added to the discussion section.

**R2:** *6. Table 1: Add also at least TPR, TPN and CALIOP cloud fraction to a table. It is the 2019 data that is used for the table, right? Add info in the caption. Include also a table with results for the unfiltered data.*

**AR:** We have replaced this table with two tables (one for the filtered dataset, and one for the unfiltered dataset). Each has BACC, TPR and TNR for all three masks and the CALIOP cloud fraction for each surface type and time of day.

**R2:** *7. Line 265: I find this surprising, I would have expected the 2D feature to be most useful for fractional clouds.*

**AR:** Quoting from our author comment: "We agree that this result is a bit counter-intuitive. We believe this is a limitation of using CALIOP as our source of labeled data. It is likely that small broken clouds are not well represented in our collocation database due to the size of these clouds and the time difference between when the two instruments observe them. Small horizontal displacements of these clouds between times that both VIIRS and CALIOP observe the same ground location could be mean that some of our labels for these types of clouds are more prone to error, and result in poor characterization of them from our neural network approach."

**R2:** *8. Line 306: You mention that Bayesian algorithms might be affected by climatological means. Considering that your method includes latitude could it not be that it too uses the latitude mean cloudiness from the two years of training data? Have you tested how much the network depends on latitude?*

**AR:** Quoting a portion of our author comment:

"Overall, the model seems to mostly worsen in nighttime water scenes when removing latitude and information related to latitude. Considering these results, I think it is likely that our model depends on latitudinal mean cloudiness in some capacity over these areas. However, it is difficult to quantify whether it is serving a purpose similar to that of a climatological mean, or if it is changing the usage of other observations features (like we have previously observed in our other models)."

We have added some details in the discussion about the neural network's dependence on latitude.

**R2:** *9. Line 357: Can the latitude combined with sun zenith angle give a rough estimate of surface temperatures? Very impressive results for temperatures close to surface temperature.*

**AR:** In our original author comment we provide a short analysis that suggest that some information in our model could give an extremely rough estimate of surface temperature. Using four predictors (solar zenith angle, the land/water mask, sun glint angle, and the absolute value of latitude) we are only able to predict GFS surface temperature with a mean absolute error of 6.7 K. and mean squared error of 86.3 K. To add an additional answer to this question we retrain our approach without the above four inputs. There are only small differences observed in our thermal contrast figure (Figure 8 of the original manuscript). As a result, we don't expect that this information plays a large role in the neural network's ability to predict cloud cover in scenes with low thermal contrast with the surface.

**R2:** *10. Line 390: I agree it is not bad with a consistent TPR dependent only on the cloud. But optimizing TPR differences might mean making the TPR lower in easy conditions to match the performance in more difficult conditions. Is it not equally important to keep TPN as constant as possible? I think this is what is more traditionally aimed at.*

**AR:** Quoting from our original author comment:

"I think the usefulness of this metric perhaps depends on the application. There is indeed a tradeoff between performance with respect to CALIOP and cloud detection consistency. For operational nowcasting, we expect that users might not be especially concerned with detection consistency over different surface types or times of day. For climate applications, this might be a more important consideration. For example, a globally uniform increase in the amount of clouds with optical depth of about 0.1 might only be detected as a much larger increase over certain surface types with higher TPR for these specific clouds. This might result depictions of cloud cover change that don't align with reality. Minimizing TNR differences between different conditions could achieve a similar result, we think. TPR differences might be more useful since one could weigh the differences with respect to optical depth. For example, a large TPR difference at a high cloud optical depth is likely more problematic than one at a very low optical depth."

We agree with referee #2's comment and have revised the text and removed the suggestion to optimize for low TPR differences. Instead, we suggest that TPR differences be used simply to identify issues related detection consistency.

**R2**: *11. Line 425: For the validation data do you have sea ice cover to the north west of Greenland? Can the shrinking sea ice cover in the arctic be part of the explanation. If MVCM is trained on older data and assumes it to be sea ice, and the new NN approach trained on more recent data expects more water?*

**AR:** We respond to this point in the individual author comment. Related to this question, we add some text mentioning how mischaracterization of surface type/temperature could be a source of error in these cloud detection models.

**R2:** *12. Line 433: The averages across space are weighted by the cosine of latitude expressed in radians. I do not understand what you mean here.*

**AR:** We have revised this sentence to more clearly state what is meant.

**R2:** *13. Line 457: The pseudo-labeling model likely has low skill in such conditions due to the low contrast between a low-level fractionally cloudy pixel and the background. Did you consider using the ECM for the pseudo-labeling?*

**AR:** Related to this comment, we add text about potentially using the MVCM and ECM predictions as pseudolabels to address deficiencies in the neural network.

**R2:** *14. Line 537: Additionally, we have not evaluated how the neural network performs specifically in cloud-free scenes with high aerosol loading. We expect that this could depend largely on the ability for CALIOP to distinguish cloud from aerosol layers. Even if it does depend on CALIOP's ability should it not depend mostly on the VIIRS capabilities?*

**AR:** Agreed. We wanted to mention the point that CALIOP's ability to distinguish cloud from aerosol would be an additional complicating factor. We have revised this statement.

**R2:** *15. Figure 6: Consider adding figures also for the BACC.*

**AR:** We have added this figure to main text.

**R2:** *16. Figure 8: Is this filtered or unfiltered data?*

**AR:** We have remade this figure with both datasets (although it was originally filtered data).

**R2:** *17. Have you tested to applid the NN on older data (2013) and was there a difference in performance?*

**AR:** Quoting from our original author comment:

> "Not yet, but we plan to examine something similar to this in future work. Whether or not ML-based approaches generalize to data collected long after (or before, in this case) their training period is an interesting question. This perhaps partially depends on whether our approach learned variability that is specific to our training period (2016 and 2018) which seems clearly undesirable. Of course, consistent sensor calibration is also a significant concern with this."

**R2:** *18. Is execution time comparable with the operational cloud masks? Is it feasible to use for nowcasting?*

**AR:** Quoting from our original author comment:

> "I wasn't particularly comfortable commenting on execution time in the manuscript since it might vary greatly across implementations or systems. Using a fairly recent GPU processing an entire 6-minute VIIRS scene with our neural network implementation took roughly 7 seconds and only hit roughly 50% utilization of the GPU. To contrast, using 2 threads of a very old (8-10 years?) CPU it took 90 seconds for the full scene and roughly 25 seconds using 12 threads. Nowcasting is certainly a possibility if the machine has access to a GPU or a modern

CPU. If running on older hardware, the processing time might be unfortunately prohibitive. That being said, we are definitely not expert software engineers, so others may have more luck in writing fast processing code.

I am not familiar with the MVCM processing code. Based only on the method itself, I suspect the MVCM is the fastest, and that the neural network is by far the slowest if run on similar hardware."

**R2:** *19. A name of the method would be useful.*

**AR:** Agreed. To help make the text more concise, we abbreviate the neural network cloud mask as NNCM.

**R2:** *20. From my experience with NN cloud masks results often look less realistic close to the swath edges when comparing results to the RGB. In Figure 12 results look realistic also closer to the edges. Is this normally the behavior?*

**AR:** Quoting from our original author comment:

"Figure 12, specifically, is a cropped image close to the nadir track of VIIRS to focus on the area of sun glint.

Early experiments we performed with MODIS/CALIOP data showed issues with viewing angle, but this is expected due to MODIS only making near-nadir collocations with CALIOP. Håkansson et al. 2018 discuss this as well with cloud-top pressure/altitude.

VIIRS makes collocations with CALIOP at a larger variety of viewing angles (0 to 50 degrees in our dataset). Early on in the development of our model we included sensor zenith angle as a predictor. However, we noticed that performance with respect to CALIOP increased when it was removed so we left it out. Aside from the regional analysis over Greenland, we have not extensively analyzed full scenes to the swath edges aside from a set of 20-30 scenes that we run as a "sanity check". In these scenes we have not noticed particularly unrealistic behavior at swath edges."

**R2:** *3 Technical corrections • Line 53: Häkansson should be Håkansson (several places)*

**AR:** This has been corrected.

**Referee #3 Comments**

**R3:** *Major Comment: What is exactly, is given in the input of the main neural network? Is it radiances, or reflectances and brightness temperature? For each radiances, is it the differences with climatology (or simulations) and the observation, or raw radiances? My concern is that the neural network description lack about the physic that is behind such the nature of the input. Also that important information are dispatch in all the study to explain fairly some results, but they are still necessary to be mentioned in the neural network description.*

**AR:** We agree that this is important information, and have added more detail in the text about the inputs into the neural network to hopefully make this more clear.

**R3:** *This is more a thought for the conclusion: How does neural network methods will react in the context of global warming and the fast modification of some surfaces? Does it mean that despite the benefits of the accuracy that provide neural network, they are countered by the fact they will need regular updates?*

**AR:** Quoting from our original author comment:

"[...] One obvious scenario, which we somewhat touch on in this paper, is the declining presence of arctic sea ice. Evidenced by Figure 10, the cloud masks in this paper can have large TPR differences over different geographical regions, time of day, and surface type. Based on Figure 2, Figure 10, and previous evidence (Liu et al., 2010), we suspect the MVCM underestimates cloud cover over sea ice. As a result, we expect that the decrease in Arctic sea ice, and corresponding increase in ice-free ocean would result in a change in cloud cover estimated by the MVCM even if it did not result in a real change in cloud cover. This is obviously undesirable, and lends us to believe that

cloud detection ability should be invariant as possible to changes in surface type. We believe that Figure 10 illustrates that the neural network may be the least susceptible to this, although the ECM and MVCM are certainly competitive with it in certain regions/conditions.

We suspect that another way in which cloud masks could be impacted by climate change would be a change in the global or regional cloud fraction. The neural network cloud mask is ultimately a statistical model. Under uncertain conditions, it will tend to predict the majority class (usually cloudy) since this is what gives it the best results during training (as measured by binary cross-entropy in our case). We have tried to ensure our approach does not depend too heavily on the use of the background mean cloud fraction by using class-balanced metrics like BACC in our evaluation. Along this same line of thinking we have limited to the amount of geographical information to the absolute value of latitude and a land/water mask. Not including longitude, or more specific surface types was an intentional choice in an effort to reduce on the model's reliance on how the mean cloud fraction varies with this information.

Regular updates would certainly be beneficial, but are not necessarily a specific requirement of our approach. If climate change were to change something fundamental about the decision boundary between clear-sky and cloudy scenes, then all statistical cloud detection models would be impacted. We hypothesize that drifts in sensor calibration or changes in noise levels would be the largest factor in whether a cloud detection model would need regular updates if used in climate data records. [...]"

We have included some discussion about how changes in surface types could impact these methods.

**R3:** *It would help to provide a table of the VIIRS band.*

**AR:** We have added this table.

**R3:** *Page 4, section 2.2: You mentioned in the discussion section (page 16, line 505) that you use ancillary data. But it is poorly described in this section 2.2 (linked to my major comment).*
**R3:** *Page 12, lines352-354: "This is surprising ... a land or water surface." This is really important information it should be mentioned in the description of the neural network input section. (major comment)*
**R3:** *Page 23, Figure 1: This paper would benefits of a better scheme that describe the neural network. Better description of the input vector with geo-localisation information.*

**AR:** We have revised some of the text describing the inputs to hopefully make this information more clearly communicated. We have also added tables detailing the VIIRS/CrIS Fusion channels and one detailing which inputs are used in each neural network model.

**R3:** *Page 6, lines 171-181: The second part of the section "3.1 Pseudo-Labelling Procedure" is hard to understand at some points. In this section, it is about the neural network that help to account for sun glint. What information is provided by this neural network to detect sun glint? Is this information provided to the main neural network to not perform a cloud mask, or does it simulate input that are supposed to appear in sun glint condition for the main neural network? Where comes from the information of true sun glint conditions, to be reproduced? Why the 15th day of every month in 2018?*

**AR:** We answer the referee's questions in our original author comment. We have revised section 3.1 in an effort address many of these questions and to make the overall presentation of these ideas more clear.

**R3:** *Page 7, line 202: what is the meaning of "binary cross-entropy"?*

**AR:** We have added the binary cross-entropy equation to the text and some additional detail about its use.

**R3:** *Page 10, lines 279- 284: Seeing the Figure 4, the difference between TPR of MVCM and the one for neural network is really small. It is most likely that their performances for low broken clouds are similar.*

**AR:** We have revised the text here and in the discussion to clarify our point about the differences between the cloud masks for broken clouds and the reliability of VIIRS/CALIOP collocations for these clouds.

**R3:** *Page 10, line 314-322: In Figure 6, the cloud mask with neural network is less sensitive to variation of latitudes.*

**AR:** We have added a comment on this to the text.

**R3:** *Page 11, line 323: "All the of the previous" A word is missing!*
**R3:** *Page 14, line 428: "is subject to a large"*

**AR:** Thanks! We have fixed these errors.

**R3:** *Page 14, line 447-page 15, line 459: I suggest you put this section and Figure 12, with the section "3.1 Pseudo-Labelling Procedure", as it makes the understanding of the pseudo-labelling more clear. Also, because this section is quiet independent of all the analysis of the neural network performances.*

**AR:** This figure has been moved to the section 3.1.

**R3:** *In relation with section 3.2 page 6 and 7, can you say more about the meaning of dropout X% between each layer?*

**AR:** We have added a short description of dropout to the text.

**R3:** *Page 34, Figure 12: There are obvious difference in the behave of the cloud mask from neural network without pseudo-label and the one with pseudo-label. The second cloud mask is more "binary" (i.e. values equal to 0 or 1) than the first one. Can you comment this result? Which neural network of figure 12, have you compared during your paper?*

**AR:** Quoting from our original author comment:

"The model that we are analyzing the performance of in the results section is Figure 12.d (the neural network with pseudo-labels). Figure 12.c is shown simply to illustrate how poorly a machine learning model can perform if we don't account for obvious deficiencies in the training dataset (such as a dataset without any sun glint examples).

On the question of the differences between the two models – that is tough to answer for a couple of reasons. First, neural networks are notoriously difficult to interpret the predictions of, and secondly, sun glint scenes are out-of-domain predictions for the neural network without pseudolabels. Said differently, it is undefined behavior for this model so attempting to interpret its predictions here is even more difficult than usual.

Of course, as we mention in the paper, the neural network without pseudolabels makes erroneously cloudy predictions throughout the entire scene. This is likely because the model without pseudolabels has likely learned to associate high visible reflectivity over water with cloudy pixels since it has never seen sun glint.

All that being said, we can speculate on some of the reasons why Fig 12.d might be more "decisive" than the model in Fig. 12.c. For Fig 12.c there are likely two competing factors: a high visible reflectivity which usually indicates cloudy pixels, and warm infrared brightness temperatures which usually indicates clear pixels. These two factors could result in uncertain conditions since these pieces of information are somewhat contradictory. Fig 12.c is likely decisive because it is making predictions in a sun glint scene where it is exclusively trained with pseudolabels. Rather, than the probabilities being accurate assessments of uncertainty with respect to the CALIOP label, the probabilities in Fig 12.d likely portray the model's ability to accurately reproduce predictions made from a model that exploits solely IR information. To that end, the cloud probability/uncertainty estimates in sun glint regions are not especially useful (which we mention in lines 531-535), but we can verify the actual predicted labels from Fig 12.d appear reasonable compared to the other operation models."

In summary, it is difficult to interpret differences in predictions between these models, particularly since these are out-of-domain predictions for one of them. Thus, our guesses at why their predictions differ are mostly speculative. We have added some text to the pseudolabeling section emphasizing that these are out-of-domain predictions for the neural network without pseudo-labels.

**Referee #4 Comments**

**R4:** *Minor comments/questions: Line 85: Before going straight to the Collocation Methodology I would recommend to add a small subsection on the VIIRS instrument/observations as well as for the two operational cloud mask swith which a lot of comparisons are done.*

**AR:** Agreed. We have added more details about the VIIRS instrument, and explained the general approaches of the two operational cloud masks.

**R4:** *Line 86: Also some more information on the CALIOP data could be provided, like what is the width of one cloud layer, time of overpass etc.?*

**AR:** Some more details about the CALIOP data have been added.

**R4:** *Line 105: Would be nice to see a global map of sampling frequency of valid collocations for the training dataset, maybe even per season, like is presented for the testing dataset (Fig. 6 b).*

**AR:** We have added a figure showing the sampling frequency and the cloud fraction from CALIOP for the training, validation, and testing dataset.

**R4:** *Line 108: Observations in form of radiances/brightness temperatures? Please provide more detail on the input for the neural network.*

**AR:** We have clarified this, added a table of VIIRS channels, and a table indicating the differences in inputs for the neural networks used in this work.

**R4***: Line 109: How are the eight categories combined?*

**AR:** Quoting from the original author comment:
   "The categories are 0=shallow ocean, 1= land, 2=coastline, 3=shallow inland water, 4= ephemeral water, 5 = deep inland water, 6 = continental/moderate ocean, 7=deep ocean.

   We combine these categories to more generally describe land/water classification. In our binary land/water mask (land=1, water=0) the land category is made up of "land" and "coastline" categories of the original mask. All other water/ocean categories of the original mask are combined into a simply a water category."

We have added text detailing how these categories are combined.

**R4***: Line 175: It is not clear to me how the sun glint scenes are labeled, on a pixel-basis? There is a reference, but some more information would be nice.*

**AR:** Quoting from the original author comment:

   "Yes. The pseudolabeling model, which is invariant to sun glint, is used to make predictions where sun glint is present. The only differences between our main cloud detection model and the pseudolabeling model are that we include the infrared VIIRS/CrIS fusion channels in the pseudolabeling model, and remove all channels with solar contributions. It similarly accepts a a 3x3 pixel patch from each channel as input, and makes a cloudy/cloud-free prediction for the center pixel. After running this model for every pixel in several scenes with sun glint, we subsample these predictions (since we had more than enough data to work with)."

We have revised this section and have tried to make it more clear how these scenes are labeled.

**R4***: Line 202: All inputs are standardized.. meaning for the 3 x 3 pixels?*

**AR:** We have clarified what we meant by this in the text.

**R4:** *Line 217: Already refer to corresponding equation numbers.*

**AR:** This has been fixed.

**R4:** *Line 314: Why not continue with BACC?*

**AR:** Quoting from our original author comment:
"We initially did not include a figure with BACC since there were grid cells where the mean CALIOP cloud fraction was particularly high, and the BACC was mostly dependent on a relatively small amount of cloud-free CALIOP collocations (the Southern Ocean, for example.) Below is the same figure with BACC, and panel (b) is replaced with the CALIOP mean cloud fraction. Similar to the ACC figure in the manuscript, these maps are calculated using the filtered dataset."

We have added a second figure with BACC and have added to the text to describe why BACC results differ significantly in some areas with ACC.

**R4:** *Line 351: How are the surface temperature from the model matched, spatially and temporally, with the measurements? Some more detail should be provided.*

**AR:** We have added detail about this in the text.

**R4:** *Line 354: are smaller than*

**AR:** We have reworded this sentence

**R4:** *Line 360/Fig 6.: The large negative difference for the grid cell in front of the coast of Namibia, could that be related to biomass burning aerosol layers?*

**AR:** Quoting from the original author comment:
"This was what we originally suspected but didn't look into it much further.

We tracked down the poor performance in this grid cell to an individual nighttime scene over the where the neural network achieved an accuracy only a 25% over a stretch of roughly 550 CALIOP collocations over the ocean near the coastline that were 97% cloud-free. To compare, the ECM had 81.5% accuracy and the MVCM has 97.8% accuracy over this same stretch.

The very poor performance of this specific scene appears to be from a number of factors. One was a processing error where the land/water mask was not reduced from the 8 categories to the binary mask (we suspect a job on our cluster was preempted or terminated early).

We checked the closest daytime overpass and adjacent regions had aerosol optical depth of roughly 0.15 to 0.3 estimated by a VIIRS data. Moderate aerosol loading was somewhat apparent in the true color images.

We think the most influential factor was what appeared to be particularly cold SSTs along the coastline indicative of upwelling. Without the visible channels in this nighttime scene, the neural network misconstrued these cold surface temperatures as cloud cover. We suspect this to be a scenario where a rough estimate of surface temperature, or some indication of coastal waters could improve the accuracy of our approach. After fixing the error in the land/water mask the accuracy over this small stretch of CALIOP collocations improves from 25% to 63% -- still trailing behind the ECM and MVCM significantly."

**R4:** R4: *Line 472: Could some (pseudo) labeling technique be useful here? Or using a larger pixel matrix than 3 x 3? Maybe combined with taking information from not only 1 CALIOP profile but from adjacent profiles as well?*

**AR:** Quoting from the original author comment:

"Sure! As another reviewer suggested we might be able to pseudolabel using the MVCM or ECM to address specific deficiencies in the neural network. For the purposes of this paper we wanted to demonstrate how the neural network could be implemented in a stand-alone capacity (assuming that neither of the operational masks were available).

We briefly experimented with a convolutional neural network that used 5x5 and 7x7 patches. These models actually performed slightly worse for broken clouds and only improved performance in more homogenous scenes (represented by the filtered dataset).

Using adjacent CALIOP profiles could be interesting, but we would have to think carefully about how this might work. We noticed in previous experiments when we trained to the 5 km CALIOP product that resulting cloud mask was extremely "smooth" and did not at all capture the fine-scale variability of scenes with broken clouds. We worry that using information from adjacent profiles might lead to a similar effect.

This might be a situation where manual labeling could be a good option. One way to do this efficiently could be to find regions of broken clouds in VIIRS images, draw a bounding box around such regions, and choose a threshold on the 11μm channel that reliably separates cloud-free from cloudy pixels (using a threshold specific to each manually selected region). This is, of course, very subjective."

**R4:** *Technical corrections Line 29: ..large amounts of training data.. Line 47: ..how a very simple.. Line 298: .. compared to the MVCM.. Line 323: All of the previous.. Line 325: ..depend on the particular.. Line 363: the distribution of the.. Line 423: .. may be a result of sea ice cover. Line 428:.. is subject to a large amount..*

**AR:** These errors have been fixed.

**List of changes**

- Added reference to ESA cloud CCI algorithm (Sus et al. 2017)
- Added details about why spectral features are not precomputed
- Added descriptions of MVCM and ECM approaches
- Replaced Frey 2019 reference with updated Frey 2020 reference for MVCM
- Added references to software used
- Added comment on relative sensitivity of 1 km and 5 km CALIOP cloud products and that MVCM may be correctly identifying clouds observed in the 5 km dataset
- Replaced the BACC table with a more comprehensive table including BACC, TPR, TNR, and CALIOP cloud fraction
    - Specified in table caption the source of data (filtered)
    - Added second table with unfiltered dataset
    - Added to text discussing unfiltered results
- Replaced Figure 1 with a table describing neural network architecture
- Added some text describing the neural network's dependence on latitude
- Removed suggestion that TPR differences be minimized during training, and added comment that it could be used to identify detection inconsistency when needed
- Added comment on how mischaracterization of surface type/temperature could be a source of error in the discussion
- Rephrased how we weight the mean cloud fraction computation (Line 433 in original manuscript)
- Added comment on the potential for using MVCM and ECM for pseudolabels where appropriate
- Added new figure for map of BACC for each mask
    - Discussed in text
- Replaced Figure 8 (thermal contrast analysis) with one that contains both filtered and unfiltered data
- Added abbreviation for the neural network as NNCM
    - Remade all figures replacing "neural network" with "NNCM" where appropriate
- Added figure of spatial and seasonal distribution of collocations
- Added text describing spatial and seasonal distribution of collocations and reasoning for selecting specific years in each dataset
- Fixed several text mentions of number of collocations to correct amount (previously was counting collocations with larger than stated time difference).
- Rephrase how we refer to MVCM to be consistent with Frey et al. 2020 – In Frey et al. 2020 it is the Continuity MODIS/VIIRS Cloud Mask not MODIS/VIIRS Continuity Cloud Mask
- Corrected multiple references to Håkansson et al. 2018 paper
- Added more detail about the inputs into the neural network (including source, units, and scaling)
- Added discussion of how changes in surface types could impact these methods.
- Added a table of VIIRS bands
    - Added reference in text to it
- Added a table of the VIIRS/CrIS Fusion Channels
    - Removed list of channels in text
    - Added references in text to it
- Many significant revisions to the text in section of 3.1
- Added equation for the binary cross-entropy cost function, and text describing its use
- Clarified expectation about difference in TPR for low broken cumulus between MVCM and NNCM
- Added comment on how the neural network is less sensitive to latitude
- Moved pseudo-labeling sun glint comparison figure to section 3.1
- Added short description of dropout
- Fixed or removed broken URLs contained in the references
- Emphasized how sun glint regions are out-of-domain for the model without pseudo-labels
- Added some details of the VIIRS instrument
- Added text explaining the approaches of the ECM and MVCM
- Added more details on the CALIOP instrument
- Added a figure of the sampling frequency for the training, validation and testing datasets

- Revised text explaining the standardization of the inputs
- Fixed equation reference to TPR
- Added text explaining how GFS surface temperatures are matched with VIIRS M15
- Rephrased analysis of thermal contrast figure to improve clarity
- Added text explaining how land/water categories are combined for binary land/water mask
- Reorganized text at the beginning of discussion to accommodate the additions suggested by reviewers
- Fixed various typos and grammatical errors, including (but not limited to):
    - Line 29: "large amounts of training data"
    - Line 47: "how a very simple"
    - Line 298: "compared to the MVCM"
    - Line 323: "All of the previous"
    - Line 325: "depend on the particular"
    - Line 363: 'the distribution of the"
    - Line 423: 'may be a result of sea ice cover"
    - Line 428: "is subject to a large"
- Revisions to many sentences to make them more concise and improve clarity
- Added reference to A-train exit to explain differences between spatial distribution of collocations between 2019 and other years

[revised manuscript text omitted]